# HYPERREP: HYPERGRAPH-BASED SELF-SUPERVISED MULTIMODAL REPRESENTATION LEARNING

## ABSTRACT

Self-supervised representation learning on multimodal data plays a pivotal role in proficiently integrating and embedding information from various sources without the need for additional labeling. Notably, the majority of existing methods overlook the complex high-order inter- and intra-modality correlations characteristic of real-world multimodal data. In this paper, we introduce HyperRep, which combines the strength of hypergraph-based modeling with a self-supervised multimodal fusion information bottleneck principle. The former captures high-order correlations using hypergraphs to represent inter- and intra-modality relations, while the latter constrains the solution space, ensuring a more effective fusion of multimodal data. Our extensive experiments on four public datasets for three downstream tasks demonstrate HyperRep's superiority, as it consistently delivers competitive results against state-of-the-art methods.

## 1 INTRODUCTION

Multimodal data, comprising various information types from diverse sources, is ubiquitous in today's data-driven world. Self-supervised representation learning for multimodal data is crucial, as it allows efficient fusion and embedding of information without requiring additional labels. This learning approach uncovers meaningful intrinsic patterns, making it ideal for various downstream applications like clustering Xu & II (2005); Xu & Tian (2015); Asano et al. (2020), text-to-video retrieval Alayrac et al. (2020); Chen et al. (2021), and temporal action localization Zhukov et al. (2019); Alwassel et al. (2020), *etc.* Effectively utilizing self-supervised multimodal representation learning can lead to more robust and versatile algorithms, addressing numerous real-world problems and advancing machine learning research.

Existing self-supervised representation learning methods for multimodal data are generally divided into pseudo-label-based Alwassel et al. (2020); Chen et al. (2021) and contrastive-based approaches Asano et al. (2020); Alayrac et al. (2020). While these methods have shown promise, they often overlook two key elements. First, they underrepresent the intricate high-order relationships inherent in multimodal data. Such correlations, like cross-modality within the same instance or cross-instance within the same modality, are integral to fully understanding the data. For example, consider a video of a car drifting. This might have high-order correlations with related images, engine sounds, and a text like "a car whizzing by", as illustrated in Fig.1(a). Similar correlations can be seen between instances of the same modality, as in Fig.1(b). Second, many existing methods lack clear principles for effective multimodal fusion, leading to potential redundancy or information loss. Addressing both these high-order relationships and fusion principles is vital for advancing representation learning in multimodal datasets.

While some methods attempt to incorporate high-order correlations in multimodal representation learning Gao et al. (2012); Zhang et al. (2018a;b), they rely on semi-supervised approaches. These require additional labeling information, which is often unavailable in many applications due to the labor-intensive nature of labeling, thus limiting their general applicability. Additionally, existing graph learning methods for multimodal representation learning Ektefaie et al. (2023) focus solely on pairwise relationships, neglecting the crucial high-order correlations that are commonly present in such data.

In this work, we present HyperRep, a pioneering approach to multimodal representation learning that masterfully bridges the intricate interplay of inter- and intra-modality correlations while ensuring that

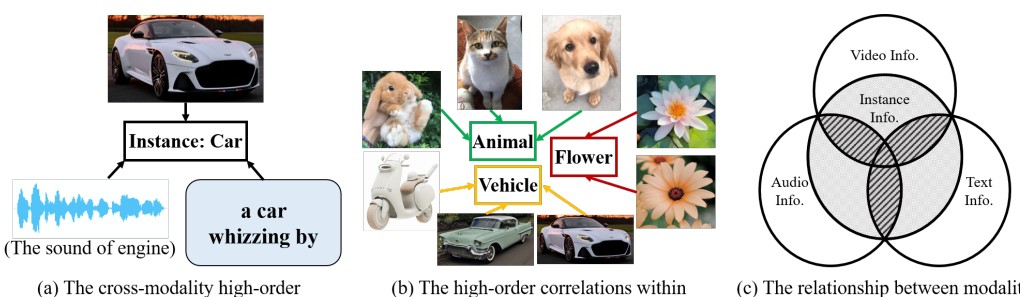

(a) The cross-modality high-order correlations within the same instance.

(b) The high-order correlations within the same modality.

(c) The relationship between modality information and instance information

Figure 1: An illustration of (a) inter-modality high-order correlations; (b) intra-modality high-order correlations; and (c) the relationship between modality-specific information and instance information.

essential information from each modality is accurately captured and retained. Central to HyperRep are two intertwined innovations: a hypergraph-based modeling technique and the self-supervised **M**ultimodal **F**usion information **B**ottleneck (MFB) principle.

Hypergraph offers a robust means to represent high-order structures resulting from intricate correlations spanning both within and across modalities. These hypergraphs, by virtue of connecting related vertices via a hyperedge, adeptly extract nuanced information that resonates across a group. We meticulously structure this representation by conceptualizing information from an individual modality of an instance as a vertex. This gives rise to two distinct hypergraph structures: the instance hypergraph, zeroing in on cross-modal instance correlations, and the modality hypergraph, tailored to hone in on cross-instance modality correlations. Such a dual hypergraph approach ensures a rich, all-encompassing capture of complex data, staving off any potential information dilution.

However, representation alone isn't enough. Introducing the MFB principle, a crucial mechanism that adeptly captures the core essence of multimodal data. Fig. 1(c) visually illustrates the fundamental concept of MFB: ensuring instances are infused with the shared modality information – the overlap where the shaded region encompasses the slashed zone. MFB plays a pivotal role by narrowing down the solution space, driving the model's gaze toward shared inter-modality information. It is not merely about contrastive learning between an instance and its modalities, but striking a fine balance by information bottleneck when faced with a huge amount of data from all modalities combined. To compute the MFB, we estimate the bounds of mutual information, allowing for an effective model optimization.

**Contributions.** In summary, our contributions are as follows: **(a)** We propose a hypergraph-based multimodal representation learning method that fully exploits high-order intra- and inter-modality correlations in multimodal data. **(b)** We introduce the self-supervised multimodal fusion information bottleneck principle to constrain the solution space and enhance the fusion of multimodal data. **(c)** Experiments are conducted on public benchmarks and achieve state-of-the-art results. Ablation studies confirm the effectiveness of each part of our proposed method.

## 2    RELATED WORK

### 2.1    SELF-SUPERVISED MULTIMODAL REPRESENTATION LEARNING

The advent of large-scale video datasets Miech et al. (2019) has fueled the evolution of representation learning approaches exploiting multimodal information in videos Zhu & Yang (2020); Sun et al. (2019); Patrick et al. (2021); Lei et al. (2021); Gabeur et al. (2020); Dong et al. (2022); Amrani et al. (2021); Alwassel et al. (2020); Asano et al. (2020); Alayrac et al. (2020); Chen et al. (2021). The key strategies involve contrastive-based and pseudo-labeling-based methods. XDC Alwassel et al. (2020), for instance, uses pseudo-labels from one modality to supervise another but yields separate representations, affecting cross-modality comparability. To mitigate this, MCN Chen et al. (2021) cultivates a joint space for multimodal data, aligning features with the same pseudo-labels. However, pseudo-labeling can generate inaccuracies and degenerate solutions. Alternatively, SeLaVi Asano et al. (2020) considers multi-modal data as instance augmentations and ensures permutation invariance, though it may dilute unique data features. Our approach is designed to

balance the benefits of contrastive methods and preserve the unique aspects of each data point, achieved through the construction of dual types of hypergraphs.

## 2.2 MULTIMODAL HYPERGRAPH LEARNING

Most multimodal learning research with hypergraphs leans towards semi-supervised approaches. For example, the MHL method Gao et al. (2012) constructs individual hypergraphs for each modality and optimizes their weights through alternating strategies. CDMH Zhang et al. (2018b) utilizes a multi-hypergraph structure to model multimodal data correlation and achieves convergence through a cross-diffusion process. Likewise, IMHL Zhang et al. (2018a) employs a multi-hypergraph to model correlations and supervises a projection from multimodal data to labels. AHGAE Hu et al. (2023), a recent unsupervised work, focuses on vertex representation for clustering, employing an adaptive hypergraph Laplacian smoothing filter and a relational reconstruction auto-encoder. However, this approach isn't explicitly tailored for multimodal data. In this work, we propose a method specifically tailored for multimodal data, using hypergraph structures to capture high-order correlations within and across different modalities.

## 3 METHOD

We are given a set of *unlabeled* multimodal data comprising $n$ instances, each containing multimodal information such as video, audio, and text. Our goal is to learn instance representations for downstream tasks. In this section, we present our HyperRep method. We begin by introducing the construction of the hypergraph structure in Section 3.1. This is followed by an explanation of the hypergraph propagation process in Section 3.2. Afterward, we describe how the self-supervised multimodal fusion information bottleneck principle is employed for optimization in Section 3.3. For readers unfamiliar with hypergraph learning, a brief introduction is provided in the Appendix A.

Basic notations and definitions are provided as follows. $n$ denotes the number of instances. The subscript $s$ refers to instance, and $v, a, t$ refers to video, audio, and text modalities, respectively. The instance set is defined as $\mathbb{S} = \{s_1, s_2, \ldots, s_n\}$. Each instance $s_i$ contains three modalities, each of which is considered as a separate vertex in this work. The vertex set is denoted as $\mathbb{V} = \mathbb{V}_v \cup \mathbb{V}_a \cup \mathbb{V}_t$, where $\mathbb{V}_v, \mathbb{V}_a$, and $\mathbb{V}_t$ represent the vertex set of video, audio, and text modality, respectively. Correspondingly, the hyperedge set is defined as $\mathbb{E} = \mathbb{E}_s \cup \mathbb{E}_m$, whereas the instance and modality hyperedge sets are defined as $\mathbb{E}_s = \{e_s^1, e_s^2, \ldots, e_s^n\}$ and $\mathbb{E}_m = \mathbb{E}_v \cup \mathbb{E}_a \cup \mathbb{E}_t$, respectively. The vertex features and hyperedge features are denoted as $X \in \mathbb{R}^{|\mathbb{V}| \times d}$ and $Y \in \mathbb{R}^{|\mathbb{E}| \times d}$, respectively, where $d$ denotes the dimension of the feature space. The incidence matrix of the whole hypergraph is defined as $H$, and $H_s$ and $H_m$ refer to the incidence matrix of the instance hypergraph and modality hypergraph, respectively.

## 3.1 HYPERGRAPH CONSTRUCTION

In the proposed model, the information from a single modality of an instance is treated as a vertex $v$. On this basis, dual types of hypergraphs are constructed: the instance hypergraph and the modality hypergraph.

**Instance hypergraph.** Different modalities in multimodal data are inherently interconnected. To capture these intrinsic correlations, we construct the instance hypergraph. Each instance contains multimodal information, and the instance hypergraph links corresponding cross-modal data. Specifically, the $i$-th instance hyperedge $e_s^i = \{v_v^i, v_a^i, v_t^i\}$, connects vertices that belong to the same instance. As shown in Fig. 2, the pink lines represent the instance hyperedges, each connecting three vertices from different modalities. The incidence matrix between the video vertex set $\mathbb{V}_v$ and instance hyperedge set $\mathbb{E}_s$ is defined as:

$$H_{s\,i,j}^v = \left\{ \begin{array}{ll} 1, & \text{if } v_v^i \in e_s^j \\ 0, & \text{if } v_v^i \notin e_s^j \end{array} \right. . \tag{1}$$

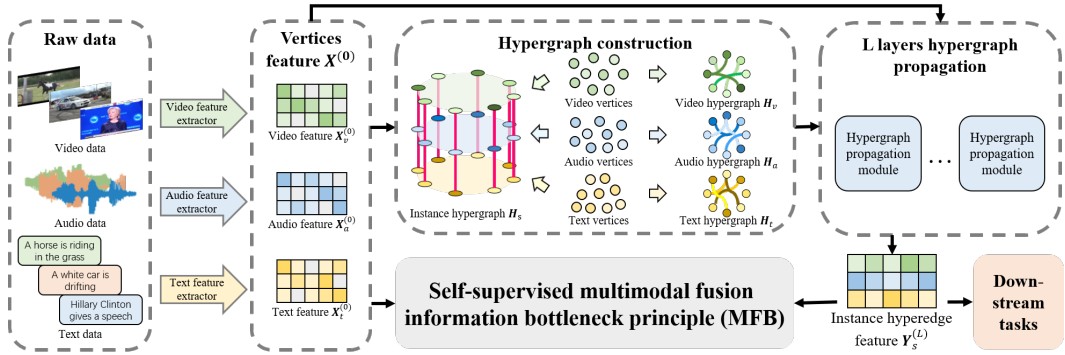

Figure 2: The pipeline of HyperRep. The hypergraph propagation module is shown in Fig. 3.

The same definition applies to audio and text modalities. Therefore, the incidence matrix between the full vertex set $\mathbb{V}$ and instance hyperedge set $\mathbb{E}_s$ can be calculated as:

$$\boldsymbol{H}_s = \begin{bmatrix} \boldsymbol{H}_s^v \\ \boldsymbol{H}_s^a \\ \boldsymbol{H}_s^t \end{bmatrix}. \tag{2}$$

**Modality hypergraphs.** The modality hypergraphs, namely the video hypergraph, audio hypergraph, and text hypergraph, capture the semantic correlations within each modality. As illustrated in Fig. 2, the video, audio, and text hyperedges are represented by green, blue, and yellow lines, respectively, with each line connecting several vertices from its corresponding modality. Hyperedges connect vertices that share similar semantics, which are identified based on the $k$-Nearest Neighbor ($k$-NN) algorithm. This approach aligns with the methodology used in HGNN Feng et al. (2019). The incidence matrix of the modality hypergraph between the video vertex set $\mathbb{V}_v$ and video hyperedge set $\mathbb{E}_v$ is given by:

$$H_m^v(i,j) = \begin{cases} 1, & \text{if } \boldsymbol{v}_v^j \in k\text{-NN}(\boldsymbol{v}_v^i) \\ 0, & \text{if } \boldsymbol{v}_v^j \notin k\text{-NN}(\boldsymbol{v}_v^i) \end{cases}. \tag{3}$$

A similar process is followed for the audio and text modalities. Thus, the incidence matrix of the modality hypergraph between the full vertex set $\mathbb{V}$ and the modality hyperedge set $\mathbb{E}_m$ is:

$$\boldsymbol{H}_m = \begin{bmatrix} \boldsymbol{H}_m^v & \boldsymbol{0} & \boldsymbol{0} \\ \boldsymbol{0} & \boldsymbol{H}_m^a & \boldsymbol{0} \\ \boldsymbol{0} & \boldsymbol{0} & \boldsymbol{H}_m^t \end{bmatrix}. \tag{4}$$

### 3.2 Hypergraph propagation

After constructing the hypergraphs, we introduce the hypergraph propagation module. In the proposed model, we utilize instance hyperedge features as instance representations for downstream tasks. This necessitates access to hyperedge features within our model. As illustrated in Fig. 3, information propagates from vertices to hyperedges and then back to vertices. Specifically, the information of vertices is aggregated to the corresponding hyperedges via the hypergraph structure to extract the high-order group features, and then passed back to the corresponding vertices. Therefore, the general paradigm of propagating information from vertex set $\mathbb{V}$ to hyperedge set $\mathbb{E}$ and back to $\mathbb{V}$ through the hypergraph structure $\boldsymbol{H}$ in the $l$-th layer is formulated as:

$$\boldsymbol{Y}^{(l+1)} = f(\boldsymbol{X}^{(l)}, \boldsymbol{Y}^{(l)}, \boldsymbol{H}), \boldsymbol{X}^{(l+1)} = f(\boldsymbol{Y}^{(l+1)}, \boldsymbol{X}^{(l)}, \boldsymbol{H}^\top), \tag{5}$$

where $\boldsymbol{X}^{(l)}$ and $\boldsymbol{Y}^{(l)}$ represent the features of vertices and hyperedges at layer $l$, and $f$ is the hypergraph propagation function.

We then define the basic version of the hypergraph propagation function $f$ as:

$$f^p(\boldsymbol{X}, \boldsymbol{H}) = \boldsymbol{D}^{-1} \boldsymbol{H}^\top \boldsymbol{X} \boldsymbol{\Theta}, \tag{6}$$

where $\boldsymbol{D} = \text{diag}(\boldsymbol{d})$ and $d_i = \sum_j H_{j,i}$, and $\boldsymbol{\Theta} \in \mathbb{R}^{d \times d}$ is the learnable parameter matrix. Consequently, $\boldsymbol{D}$ represents the edge degree matrix $\boldsymbol{D}_e$ and vertex degree matrix $\boldsymbol{D}_v$ when the input is $\boldsymbol{H}$

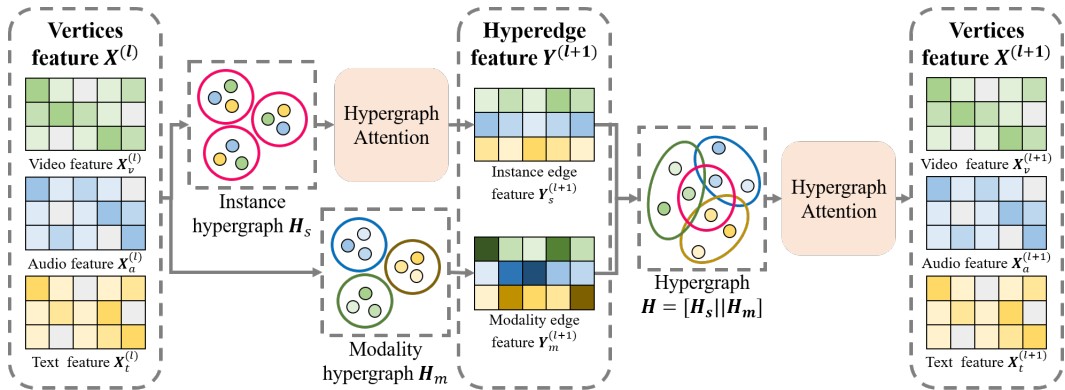

Figure 3: Hypergraph propagation module described in Eq. 8 to Eq. 11. Unlike in Fig. 2, hyperedges are represented as circles here for the sake of clarity. However, their colors remain consistent.

and $\boldsymbol{H}^\top$, respectively. However, cross-modality information may have varying importance in each instance. To overcome this limitation, we utilize the hypergraph attention module to achieve better fusion across different modalities.

**Hypergraph attention module.** The attention mechanism within the hypergraph is designed to learn the attention weights between vertices and hyperedges. This is because different vertices have varying degrees of importance for the corresponding hyperedges, and vice versa. Consequently, we perform the scaled dot-product attention Vaswani et al. (2017) from vertices to hyperedges with mask $\boldsymbol{H}$, and define the propagation function $f$ as:

$$f^{attn}(\boldsymbol{X}, \boldsymbol{Y}, \boldsymbol{H}) = \text{Softmax}(\text{Mask}(\frac{\boldsymbol{Y}\boldsymbol{W}^q(\boldsymbol{X}\boldsymbol{W}^k)^\top}{\sqrt{\boldsymbol{d}_k}}, \boldsymbol{H}^\top))\boldsymbol{X}\boldsymbol{W}^v, \tag{7}$$

where $\boldsymbol{W}^q \in \mathbb{R}^{d \times d_k}$, $\boldsymbol{W}^k \in \mathbb{R}^{d \times d_k}$, and $\boldsymbol{W}^v \in \mathbb{R}^{d \times d}$ are learnable parameter matrices, and $\frac{1}{\sqrt{\boldsymbol{d}_k}}$ is the scaling factor. In essence, attention weights are considered only between the vertices and the hyperedges that are associated with the incidence matrix $\boldsymbol{H}$ created previously. Through the use of the hypergraph attention module, attention-weighted aggregation information can be obtained.

**Propagation process.** The hypergraph propagation module operates as follows:

$$\boldsymbol{Y}_s^{(0)} = \frac{1}{3}(\boldsymbol{X}_v^{(0)} + \boldsymbol{X}_a^{(0)} + \boldsymbol{X}_t^{(0)}), \tag{8}$$

$$\boldsymbol{Y}_s^{(l+1)} = f^{attn}(\boldsymbol{X}^{(l)}, \boldsymbol{Y}_s^{(l)}, \boldsymbol{H}_s^\top), \tag{9}$$

$$\boldsymbol{Y}_m^{(l+1)} = f^p(\boldsymbol{X}^{(l)}, \boldsymbol{H}_m^\top), \tag{10}$$

$$\boldsymbol{X}^{(l+1)} = f^{attn}([\boldsymbol{Y}_s^{(l+1)}\|\boldsymbol{Y}_m^{(l+1)}], \boldsymbol{X}^{(l)}, [\boldsymbol{H}_s\|\boldsymbol{H}_m]), \tag{11}$$

where $\cdot\|\cdot$ denotes concatenation operation. After propagation through $L$ layers, the instance hyperedge feature $\boldsymbol{Y}_s^{(l+1)}$ is used for the execution of downstream tasks.

The hypergraph propagation module we've designed serves a dual purpose: extracting cross-modal instance semantic consistency and modality-specific semantics from different hypergraphs. Simultaneously, it ensures the intricate data from these varying hypergraphs is preserved by the vertices, minimizing the potential for significant information loss.

### 3.3 SELF-SUPERVISED MULTIMODAL FUSION INFORMATION BOTTLENECK PRINCIPLE

Multimodal representations encapsulate both the shared information across modalities and the unique feature information specific to each modality. As depicted in Fig. 1 (c), individual circles represent the information of a single modality, while the shaded circle symbolizes the information of the instance. The overlapped, slashed portions of the modality circles represent the shared information jointly expressed across two or three modalities. In contrast, the distinct white sections denote the modality-specific feature information.

Ideally, the instance information should contain the modality-shared information, *i.e.*, the shaded area of the instance circle should encompass the slashed areas shared by the modality circles. To achieve this, we introduce the **M**ultimodal **F**usion information **B**ottleneck (MFB) principle, which aims to maximize the mutual information between the instance and each modality, while minimizing the mutual information between the instance and the totality of information. In terms of Fig. 1 (c), this can be viewed as maximizing the area of overlap between the instance circle and each modality circle, while minimizing the overlap between the instance circle and the union of all modality circles. By guiding the instance representation learning process to focus more on the shared multimodal information, MFB effectively constrains the solution space to a narrower range, directing the model's attention towards the shared information across modalities.

The MFB principle for the $l$-th layer instance hyperedge representation is formulated as follows:

$$\min_{p(\boldsymbol{Y}_s^{(l)}|\boldsymbol{X}^{(0)})\in\Omega} \text{MFB}(\boldsymbol{Y}_s^{(l)};\boldsymbol{X}^{(0)}) \triangleq -\sum_m \mathcal{I}(\boldsymbol{X}_m^{(0)};\boldsymbol{Y}_s^{(l)}) + \beta\mathcal{I}(\boldsymbol{X}^{(0)};\boldsymbol{Y}_s^{(l)}), \qquad (12)$$

where $\Omega$ represents the search space of the conditional distribution of $\boldsymbol{Y}_s^{(l)}$ given the initial vertex feature $\boldsymbol{X}^{(0)}$, and the hyper-parameter $\beta$ serves to balance the weight of the two components.

**Estimation of MFB.** Since mutual information becomes intractable when the probability distribution is unknown, we perform upper and lower bound estimations to enable its computation and training via back propagation.

**Proposition 1.** *The upper and lower bounds of the mutual information between two random variables* **x** *and* **y** *can be estimated as:*

$$\mathbb{E}[\log \frac{f(\boldsymbol{y}_+,\boldsymbol{x})}{\sum_{vy_i\in Y} f(\boldsymbol{y}_i,\boldsymbol{x})}] \le \mathcal{I}(\mathbf{x};\mathbf{y}) \le D_{\text{KL}}(p(\mathbf{x}|\mathbf{y})\|q(\mathbf{x})), \qquad (13)$$

*where* $\boldsymbol{x}$ *and* $\boldsymbol{y}_+$ *are positive pairs sampled from* $p(\mathbf{x}|\mathbf{y})$, $f(\cdot,\cdot)$ *is a scoring function that measures the similarity between two embeddings, and* $q$ *is a prior distribution of* **x**.

The proof is provided in the Appendix B. The form of the mutual information's lower bound above is known as the InfoNCE loss van den Oord et al. (2018). Consequently, the MFB loss can be expressed as:

$$\mathcal{L}_{\text{MFB}} = \sum_m \mathcal{L}_{\text{InfoNCE}}(\boldsymbol{X}_m^{(0)},\boldsymbol{Y}_s^{(l)}) + \beta D_{\text{KL}}(p(\boldsymbol{Y}_s^{(l)}|\boldsymbol{X}^{(0)})\|q(\boldsymbol{Y}_s^{(l)})). \qquad (14)$$

The calculation of the MFB loss is detailed in the Appendix C.

## 4 EXPERIMENTS

In order to evaluate the quality of the representations learned by HyperRep, we conduct a series of experiments on downstream tasks. These experiments encompass three primary areas of investigation: (1) comparisons against state-of-the-art methods on clustering task in Section 4.1; (2) ablation studies for each component of the proposed method in Section 4.2; (3) more downstream tasks, including text-to-video retrieval and temperoal action localization task, in Section 4.3. The implementation details, sensitivity and convergence analysis, computation complexity analysis can be found in the Appendix E, J and I, respectively.

### 4.1 EXPERIMENTS ON CLUSTERING TASK

**Datasets.** We perform clustering experiments on three publicly available datasets: AVE (Audio-Visual Event) Tian et al. (2018), MSR-VTT (Microsoft Research Video to Text) Xu et al. (2016), and YouCook2 Zhou et al. (2018). The detailed description of datasets can be found in the Appendix D. Each of these is a video dataset from which we extract multimodal information. We filter out instances with missing modalities.

**Metrics.** We assess our method using the metrics of *accuracy* (Acc), *normalized mutual information* (NMI), and *adjusted rand index* (ARI). The computation of metrics can be found in the Appendix F. The *accuracy* is calculated post self-supervised label matching to the ground truth via the Kuhn-Munkres algorithm Kuhn (1955).

Table 1: Experimental results compared with state-of-the-art methods. The best performance is highlighted in bold, and the second-best performance is underlined.

| Dataset | AVE | | | MSR-VTT | | | YouCook2 | | |
|---|---|---|---|---|---|---|---|---|---|
| Model | Acc | NMI | ARI | Acc | NMI | ARI | Acc | NMI | ARI |
| K-means | $46.2 \pm 1.3$ | $54.7 \pm 0.6$ | $32.6 \pm 0.6$ | $30.7 \pm 1.3$ | $24.6 \pm 0.7$ | $13.6 \pm 1.1$ | $19.1 \pm 0.8$ | $45.0 \pm 0.6$ | $5.6 \pm 0.5$ |
| Spectral | $49.3 \pm 1.3$ | $55.6 \pm 0.6$ | $36.7 \pm 1.3$ | $34.6 \pm 0.5$ | $26.2 \pm 0.3$ | $18.1 \pm 0.2$ | $19.8 \pm 0.7$ | $46.5 \pm 0.5$ | $6.6 \pm 0.5$ |
| AGC | $63.1 \pm 0.4$ | $\underline{70.8 \pm 0.1}$ | $52.0 \pm 0.4$ | $36.4 \pm 0.5$ | $33.1 \pm 0.1$ | $16.9 \pm 0.5$ | $20.5 \pm 0.6$ | $46.8 \pm 0.6$ | $6.9 \pm 0.5$ |
| AGE | $33.0 \pm 0.1$ | $63.7 \pm 0.2$ | $26.7 \pm 0.2$ | $35.1 \pm 8.2$ | $29.3 \pm 6.1$ | $18.9 \pm 5.8$ | $6.7 \pm 1.0$ | $20.1 \pm 2.9$ | $1.3 \pm 0.4$ |
| AdaGAE | $35.5 \pm 2.3$ | $51.4 \pm 3.2$ | $22.7 \pm 3.7$ | $19.4 \pm 1.4$ | $14.2 \pm 0.7$ | $6.4 \pm 0.9$ | $22.2 \pm 0.5$ | $48.7 \pm 0.3$ | $8.0 \pm 0.1$ |
| AHGAE | $12.5 \pm 0.8$ | $36.0 \pm 2.1$ | $8.2 \pm 0.8$ | $36.9 \pm 2.9$ | $29.9 \pm 1.9$ | $21.1 \pm 3.8$ | $6.8 \pm 0.7$ | $20.1 \pm 1.9$ | $1.3 \pm 0.2$ |
| SeLaVi | $57.9$ | $66.2$ | $47.4$ | $25.1$ | $19.9$ | $9.9$ | $8.8$ | $29.5$ | $0.4$ |
| MCN | $55.9 \pm 3.1$ | $67.5 \pm 1.3$ | $45.5 \pm 2.3$ | $\underline{40.2 \pm 1.0}$ | $\underline{36.7 \pm 0.5}$ | $\underline{26.5 \pm 1.7}$ | $26.8 \pm 0.4$ | $\underline{55.6 \pm 0.6}$ | $13.0 \pm 0.4$ |
| MFLVC | $59.4 \pm 1.4$ | $70.1 \pm 1.0$ | $51.0 \pm 1.5$ | $30.1 \pm 1.3$ | $27.7 \pm 0.7$ | $16.0 \pm 1.4$ | $9.9 \pm 0.5$ | $34 \pm 1.0$ | $0.8 \pm 0.5$ |
| CrossCLR | $\underline{65.9 \pm 1.3}$ | $70.1 \pm 1.1$ | $\underline{54.3 \pm 1.6}$ | $38.0 \pm 1.2$ | $32.5 \pm 0.8$ | $22.4 \pm 1.5$ | $\underline{28.0 \pm 0.7}$ | $54.9 \pm 0.8$ | $\underline{13.5 \pm 0.7}$ |
| HyperRep | $\mathbf{68.3 \pm 2.3}$ | $\mathbf{75.7 \pm 1.1}$ | $\mathbf{60.7 \pm 2.0}$ | $\mathbf{41.8 \pm 0.5}$ | $\mathbf{37.0 \pm 0.3}$ | $\mathbf{28.8 \pm 1.0}$ | $\mathbf{29.6 \pm 1.1}$ | $\mathbf{56.9 \pm 0.9}$ | $\mathbf{16.3 \pm 1.0}$ |

**Baselines.**    We evaluate our approach against ten distinct methodologies, which fall under the following categories: (1) Feature-dependent clustering techniques, such as K-means and spectral clustering. (2) Graph and hypergraph-driven representation and clustering approaches, exemplified by AGC Zhang et al. (2019), AGE Cui et al. (2020), AdaGAE Li et al. (2022), and AHGAE Hu et al. (2023). (3) Cutting-edge video representation techniques like SeLaVi Asano et al. (2020), MCN Chen et al. (2021), MFLVC Xu et al. (2022), and CrossCLR Zolfaghari et al. (2021). It's noteworthy that SeLaVi is limited to audio and video modalities. For an equitable evaluation, models, specifically SeLaVi and MCN, which come pre-trained on other datasets, are fine-tuned during our experimentation. With the exception of SeLaVi that operates directly on raw videos, all other methodologies leverage identical input features as our approach. Although CrossCLR's primary novelty is its loss function, we match CrossCLR's performance merely by substituting the MFB loss with CrossCLR's, overlooking variations attributed to network design.

**Experimental results on clustering task.**    As displayed in Table 1, HyperRep demonstrates excellent performance across all three datasets, outperforming all other methods in all metrics. On the AVE dataset, it leads AGC by 8.2%, 6.9%, and 16.7% in Acc, NMI, and ARI metrics respectively, and outpaces CrossCLR by 3.6%, 8.0%, and 11.8%. For the MSR-VTT dataset, it surpasses MCN with margins of 4.0%, 0.8%, and 2.3%. On the YouCook2 dataset, the advantages against MCN are 10.4%, 2.3%, and 25.4%, and when compared to CrossCLR, they stand at 5.7%, 3.6%, and 20.7% for the same metrics. The consistent outperformance of HyperRep showcases its efficacy and robustness in multimodal representation learning.

Specifically, **the high Acc** shows our model's ability to accurately group instances into the correct clusters. This indicates that the multimodal representations learned by HyperRep effectively capture the specific characteristics of each instance. This accuracy suggests that the model can derive distinct representations that clearly separate instances based on their inherent attributes. **The significant ARI**, which measures the consistency between true and predicted cluster assignments while accounting for random groupings, shows that our model's representations capture the genuine similarities and differences among instances. The model's proficiency in individual instance assignment (as shown by Acc) and its capability to determine if pairs of instances should be in the same or different clusters (as indicated by ARI) emphasize the depth and quality of HyperRep's representations. Moreover, **the strong NMI** result indicates HyperRep's ability to understand the overall clustering structure. A high NMI suggests that our model is not only good at representation learning but also effectively retains the general structure and distribution of data clusters. In summary, HyperRep performs well in both representation learning and clustering tasks.

## 4.2    Ablation studies

To better understand the contributions of various components in our proposed HyperRep model, we conduct ablation studies as shown in Table 3. By removing each component in turn and observing the resulting performance, we can estimate the impact of each component on the overall effectiveness of the model. Due to space limitations, two additional ablation studies are presented in Appendix H.

Table 3: Experiment results of ablation studies.

| Dataset | AVE | | | MSR-VTT | | | YouCook2 | | |
|---|---|---|---|---|---|---|---|---|---|
| Ablations | **Acc** | **NMI** | **ARI** | **Acc** | **NMI** | **ARI** | **Acc** | **NMI** | **ARI** |
| w/o high-order corr. | $11.1 \pm 1.7$ | $22.5 \pm 8.1$ | $4.2 \pm 1.0$ | $22.9 \pm 1.2$ | $18.2 \pm 2.1$ | $9.8 \pm 0.8$ | $15.6 \pm 1.9$ | $43.5 \pm 2.7$ | $4.0 \pm 1.3$ |
| $\mathcal{L}_{\text{InfoNCE}}$ only | $67.7 \pm 2.0$ | $74.9 \pm 0.3$ | $60.4 \pm 0.9$ | $40.8 \pm 0.5$ | $36.8 \pm 0.3$ | $26.9 \pm 0.9$ | $29.1 \pm 0.2$ | $56.0 \pm 0.5$ | $15.6 \pm 0.2$ |
| Video + audio | - | - | - | $39.6 \pm 0.3$ | $35.0 \pm 0.4$ | $26.0 \pm 0.4$ | $21.0 \pm 0.3$ | $48.6 \pm 0.4$ | $7.7 \pm 0.1$ |
| Video + text | - | - | - | $36.9 \pm 1.5$ | $34.4 \pm 0.3$ | $22.0 \pm 1.6$ | $23.2 \pm 0.1$ | $50.9 \pm 0.2$ | $10.3 \pm 0.3$ |
| Audio + text | - | - | - | $41.2 \pm 1.0$ | $35.4 \pm 0.6$ | $26.6 \pm 1.5$ | $25.5 \pm 0.2$ | $53.0 \pm 0.1$ | $11.9 \pm 0.1$ |
| full model | $68.3 \pm 2.3$ | $75.7 \pm 1.1$ | $60.7 \pm 2.0$ | $41.8 \pm 0.5$ | $37.0 \pm 0.3$ | $28.8 \pm 1.0$ | $29.6 \pm 1.1$ | $56.9 \pm 0.9$ | $16.3 \pm 1.0$ |

**Ablation study of high-order correlations.** We substitute the modality hypergraph incidence matrix with the identity matrix. This modification transforms the process of propagating information from vertices to modality hyperedges into a linear layer operation. Consequently, the high-order structure intrinsic to each modality is ablated. However, we cannot ablate the instance hypergraph, i.e., the high-order cross-modality correlations, because the instance hyperedge representations are necessary for the clustering task. As depicted in Table 3, the full model outperforms this ablation by an average of 699%, 126.6%, and 142.7% on AVE, MSR-VTT, and YouCook2, respectively. The removal of high-order correlations within modalities negatively affects the model's performance. This suggests that these high-order correlations play a crucial role in multimodal learning.

**Ablation study of MFB loss.** We modify the MFB loss function to become equivalent to the InfoNCE loss by setting the hyper-parameter $\beta = 0$ in Eq. 14. Therefore, the model is optimized solely by maximizing the mutual information within each modality, without the constraint of focusing on cross-modal shared information. This implies that the learned representations could be influenced by modality-specific, instance-irrelevant features. The experimental results support this claim. The full model outperforms this ablation by an average of 0.8%, 3.4%, and 2.6% on the AVE, MSR-VTT, and YouCook2 datasets, respectively. This suggests that constraining the solution space of the representation helps focus on cross-modal shared information, thus enhancing performance.

Table 2: Comparison of text-to-video retrieval systems on the MSR-VTT dataset. The modalities are represented by V for video, A for audio, and T for text. TR indicates if a trainable backbone is used or not.

| Method | Modality | Model | TR | R@1 | R@5 | R@10 |
|---|---|---|---|---|---|---|
| Random | - | - | - | 0.01 | 0.05 | 0.1 |
| Miech | VT | R152+RX101 | N | 7.2 | 19.2 | 28.0 |
| MDR | VT | R152+RX101 | N | 8.0 | 21.3 | 29.3 |
| MIL-NCE* | VT | R152+RX101 | N | 8.4 | 23.2 | 32.4 |
| MCN | VAT | R152+RX101 | N | 10.5 | 25.2 | 33.8 |
| MDR | VT | R152 | N | 8.4 | 22.0 | 30.4 |
| ActBERT | VT | R101+Res3D | N | 8.6 | 23.4 | 33.1 |
| SSB | VT | R(2+1)D-34+R152 | N | 8.7 | 23.0 | 31.1 |
| MMV FAC | VAT | TSM-50x2 | Y | 9.3 | 23.0 | 31.1 |
| MIL-NCE | VT | I3D-G | Y | 9.4 | 22.2 | 30.0 |
| MIL-NCE | VT | S3D-G | Y | 9.9 | 24.0 | 32.4 |
| HyperRep | VAT | R152+RX101 | N | **11.6** | **26.3** | **37.3** |

**Ablation study of modality.** Lastly, we perform ablation experiments by omitting each modality in turn. Given that our method requires multimodal input, we cannot carry out this ablation on the AVE dataset, which only comprises two modalities. Instead, we exclude the video, audio, and text modalities individually on the MSR-VTT and YouCook2 datasets. The results consistently demonstrate that performance improves when all three modalities are included, as compared to when only two are used, indicating that each modality contributes significantly.

## 4.3 EXPERIMENTS ON MORE DOWNSTREAM TASKS

In this section, we provide more experiments to demonstrate the adaptability and scalability of HyperRep across various downstream tasks.

### 4.3.1 EXPERIMENTS ON TEXT-TO-VIDEO RETRIEVAL TASK

**Dataset and metric.** We conduct text-to-video retrieval experiments on the MSR-VTT (Microsoft Research Video to Text) dataset Xu et al. (2016). The primary objective is to identify videos that best match a given text description. To evaluate performance, we employ the Recall@k metric, which measures whether the target video appears within the top-k most similar videos for a given text. Implementation details is provided in Appendix G.

**Baselines.** Following MCN Chen et al. (2021), we evaluate our approach against seven state-of-the-art method, which are Miech Miech et al. (2019), MDR Amrani et al. (2021), MIL-NCE Miech et al. (2020), ActBERT Zhu & Yang (2020), SSB Patrick et al. (2021), MMV FAC Alayrac et al. (2020) and MCN Chen et al. (2021). The duplicate methods in the table use different backbones.

**Experimental results.** As illustrated in Table 2, our approach consistently outperforms all other state-of-the-art methods. When compared to the second-best method, MCN, we observe improvements of 10.5%, 4.4%, and 10.4% in Recall@1, Recall@5, and Recall@10, respectively. These performance gains are attest to the efficacy of our multi-modal representation learning approach. By bridging the semantic gap between different modalities, our method ensures that the learned representations encapsulate richer and more comprehensive information. This nuanced understanding is evident as our approach excels at aligning textual descriptions with their corresponding video narratives—a critical capability in real-world applications where users use textual queries to search for relevant video content. Furthermore, the significant lead in Recall@1 underscores our model's precision in identifying the most pertinent video based on a textual description. Such accuracy in retrieval tasks emphasizes the superiority and robustness of the multi-modal representations we've learned, which subsequently enhances user satisfaction in retrieval systems. The qualitative analysis is provided in Appendix K.

### 4.3.2 EXPERIMENTS ON TEMPORAL ACTION LOCALIZATION TASK

**Dataset and metric** We perform temporal action localization experiments using the CrossTask dataset Zhukov et al. (2019). Each video is segmented into a series of 1-second clips and is accompanied by an unordered set of action labels. The challenge lies in accurately associating each clip with its corresponding action label. The effectiveness of the model is quantified using Recall, which is calculated as the proportion of clips correctly labeled out of the total number of clips in the video. Implementation details is provided in Appendix G.

**Baselines.** Following MCN Chen et al. (2021), we evaluate our approach against five state-of-the-art method, which are CrossTask Zhukov et al. (2019), Miech Miech et al. (2019), MIL-NCE Miech et al. (2020), ActBERT Zhu & Yang (2020), and MCN Chen et al. (2021). The duplicate methods in the table use different backbones.

Table 4: Comparison of temporal action localization systems on the CrossTask dataset.

| Method | Modality | Model | TR | Recall |
|---|---|---|---|---|
| CrossTask | VT | R152+I3D | N | 31.6 |
| Miech | VT | R152+RX101 | N | 33.6 |
| MIL-NCE* | VT | R152+RX101 | N | 33.2 |
| MCN | VAT | R152+RX101 | N | 35.1 |
| ActBERT | VT | R101+Res3D | N | 37.1 |
| ActBERT | VT | + Faster R-CNN | N | 41.4 |
| MIL-NCE | VT | I3D-G | Y | 36.4 |
| MIL-NCE | VT | S3D-G | Y | 40.5 |
| HyperRep | VAT | R152+I3D | N | **50.68** |

**Experimental results.** Critically analyzing the results presented in Table 4, our method has set a new benchmark in performance. Notably, we exceed the second-best performance of ActBERT by 22.4% in Recall. This is particularly impressive given that ActBERT utilizes additional feature modalities and a more advanced language model, while we predominantly draw from the standard features provided by CrossTask. It emphasizes the ability of HyperRep to unearth and exploit the latent semantic structures across modalities.

## 5 CONCLUSION

In this study, we proposed HyperRep, a hypergraph-based method for self-supervised multimodal representation learning. Our model consistently outperformed state-of-the-art methods across all metrics and datasets, highlighting its proficiency in learning distinct and meaningful representations. The ablation studies further underlined the significance of high-order correlations, the multimodal fusion information bottleneck constraints, and the valuable contribution of each modality in multimodal learning. Moving forward, we believe that the foundational principles of HyperRep can be extended to a broader range of multimodal applications, setting a new benchmark for future research in this domain.

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

## A  HYPERGRAPH PRELIMINARY

Typically, a hypergraph can be defined as $\mathcal{G} = \{\mathbb{V}, \mathbb{E}\}$, where $\mathbb{V}$ and $\mathbb{E}$ denote the the sets of vertices and hyperedges, respectively. A hyperedge $e$ is a non-empty subset of $\mathbb{V}$ that contains multiple vertices. It denotes an interaction in which one or more vertices can participate. The incidence matrix of a hypergraph is represented as $\boldsymbol{H} \in \{0, 1\}^{|\mathbb{V}| \times |\mathbb{E}|}$, which characterizes the interactions between the vertex set $\mathbb{V}$ and the hyperedge set $\mathbb{E}$. Each entry $\boldsymbol{H}(v, e)$ indicates whether the vertex $v$ belongs to the hyperedge $e$:

$$H_{v,e} = \begin{cases} 1, & \text{if } v \in e \\ 0, & \text{if } v \notin e \end{cases}. \tag{15}$$

The degree of each vertex $v$ in a hypergraph $\mathcal{G}$ is defined as $d(v) = \sum_{e \in \mathbb{E}} H_{v,e}$, and the degree of each hyperedge $e$ is defined as $\delta(e) = \sum_{v \in \mathbb{V}} H_{v,e}$. Additionally, $\boldsymbol{D}_v \in \mathbb{N}^{|\mathbb{V}| \times |\mathbb{V}|}$ and $\boldsymbol{D}_e \in \mathbb{N}^{|\mathbb{E}| \times |\mathbb{E}|}$ represent the diagonal matrices of the vertex and hyperedge degrees, respectively.

The laplacian matrix of the hypergraph Zhou et al. (2006) is defined as:

$$\Delta = \boldsymbol{I} - \boldsymbol{D}_v^{-1/2} \boldsymbol{H} \boldsymbol{D}_e^{-1} \boldsymbol{H}^\top \boldsymbol{D}_v^{-1/2}. \tag{16}$$

Furthermore, the hypergraph convolution Feng et al. (2019) on the spectral domain is parameterized as:

$$\boldsymbol{X}^{(l+1)} = \boldsymbol{D}_v^{-1/2} \boldsymbol{H} \boldsymbol{D}_e^{-1} \boldsymbol{H}^\top \boldsymbol{D}_v^{-1/2} \boldsymbol{X}^{(l)} \boldsymbol{\Theta}^{(l)}, \tag{17}$$

where $\boldsymbol{X}^{(l)}$ and $\boldsymbol{\Theta}^{(l)}$ are the vertex feature and learnable parameter matrices at layer $l$, respectively. Motivated by the hyper-path in hypergraph, the spatial-based convolution on hypergraphs named HGNNConv$^+$ Gao et al. (2022) is defined as:

$$\boldsymbol{X}^{(l+1)} = \boldsymbol{D}_v^{-1} \boldsymbol{H} \boldsymbol{D}_e^{-1} \boldsymbol{H}^\top \boldsymbol{X}^{(l)} \boldsymbol{\Theta}^{(l)}, \tag{18}$$

where $\boldsymbol{X}^{(l)}$ and $\boldsymbol{\Theta}^{(l)}$ are also the vertex feature and learnable parameter matrices at layer $l$, respectively.

However, as shown in Eq. 17 and Eq. 18, both spectral and spatial hypergraph convolutional layers do not have access to hyperedge features. Instead, it integrates the vertex-hyperedge-vertex transformation into vertex-vertex form. This does not meet the requirements of our model, which needs the instance hyperedge representation for downstream tasks.

## B  PROOF OF PROPOSITION 1

*Proof.* The proof of mutual information's lower bound estimation can be found in the appendix of previous work van den Oord et al. (2018). Here we present the proof for the upper bound estimation. We know that the KL divergence is always greater than zero, and therefore we have:

$$D_{\mathrm{KL}}(p(\mathbf{x})||q(\mathbf{x})) = \mathbb{E}_{p(\mathbf{x})}[\log p(\mathbf{x})] - \mathbb{E}_{p(\mathbf{x})}[\log q(\mathbf{x})] \geq 0. \tag{19}$$

By following the definition of mutual information, we get:

$$\mathcal{I}(\mathbf{x}; \mathbf{y}) = \mathbb{E}_{p(\mathbf{x},\mathbf{y})}[\log \frac{p(\mathbf{x}|\mathbf{y})}{p(\mathbf{x})}] \tag{20}$$

$$\approx \mathbb{E}_{p(\mathbf{x}|\mathbf{y})}[\log \frac{p(\mathbf{x}|\mathbf{y})}{p(\mathbf{x})}] \tag{21}$$

$$\leq \mathbb{E}_{p(\mathbf{x}|\mathbf{y})}[\log \frac{p(\mathbf{x}|\mathbf{y})}{q(\mathbf{x}|\mathbf{y})}] \tag{22}$$

$$= D_{\mathrm{KL}}(p(\mathbf{x}|\mathbf{y})||q(\mathbf{x})). \tag{23}$$

Thus, we conclude:

$$\mathcal{I}(\mathbf{x}; \mathbf{y}) \leq D_{\mathrm{KL}}(p(\mathbf{x}|\mathbf{y})||q(\mathbf{x})). \tag{24}$$

$\square$

## C    THE CALCULATION OF MFB LOSS

We first give a lemma and a proposition to present the calculation of MFB loss, which is defined as:

$$\mathcal{L}_{\text{MFB}} = \sum_m \mathcal{L}_{\text{InfoNCE}}(\boldsymbol{X}_m^{(0)}, \boldsymbol{Y}_s^{(l)}) + \beta D_{\text{KL}}(p(\boldsymbol{Y}_s^{(l)}|\boldsymbol{X}^{(0)})||q(\boldsymbol{Y}_s^{(l)})). \tag{25}$$

**Lemma 1.** *Given two $J$-dimensional Gaussian distributions $p(\boldsymbol{x}) \sim \mathcal{N}_1(\boldsymbol{\mu}_1, \boldsymbol{\sigma}_1^2)$ and $q(\boldsymbol{x}) \sim \mathcal{N}_2(\boldsymbol{\mu}_2, \boldsymbol{\sigma}_2^2)$, we have*

$$\int p(\boldsymbol{x}) \log q(\boldsymbol{x}) \mathrm{d}\boldsymbol{x} = -\frac{1}{2} \sum_{i=1}^{J} [\log 2\pi + \log {\sigma_2^i}^2 + \frac{(\mu_1^i - \mu_2^i)^2 + {\sigma_1^i}^2}{{\sigma_2^i}^2}], \tag{26}$$

*where $\mu^i$ and $\sigma^i$ denote the $i$-th element of $\boldsymbol{\mu}$ and $\boldsymbol{\sigma}$, respectively.*

*Proof.*

$$\int p(\boldsymbol{x}) \log q(\boldsymbol{x}) \mathrm{d}\boldsymbol{x} = \int \mathcal{N}(\boldsymbol{x}; \boldsymbol{\mu}_1, \boldsymbol{\sigma}_1^2) \log \mathcal{N}(\boldsymbol{x}; \boldsymbol{\mu}_2, \boldsymbol{\sigma}_2^2) \mathrm{d}\boldsymbol{x} \tag{27}$$

$$= \sum_{i=1}^{J} \int \mathcal{N}_1(x_i; \mu_1^i, {\sigma_1^i}^2) \log \mathcal{N}_2(x_i; \mu_2^i, {\sigma_2^i}^2) \mathrm{d}x_i \tag{28}$$

$$= \sum_{i=1}^{J} \int \mathcal{N}_1(x_i; \mu_1^i, {\sigma_1^i}^2) \log[\frac{1}{\sqrt{2\pi {\sigma_2^i}^2}} \exp(-\frac{(x_i - \mu_2^i)^2}{2\sigma_2^i})] \mathrm{d}x_i \tag{29}$$

$$= \sum_{i=1}^{J} -\frac{1}{2} \log(2\pi {\sigma_2^i}^2) \int \mathcal{N}_1(x_i; \mu_1^i, {\sigma_1^i}^2) \mathrm{d}x_i \tag{30}$$

$$- \frac{1}{2{\sigma_2^i}^2} \int (x_i - \mu_2^i)^2 \mathcal{N}_1(x_i; \mu_1^i, {\sigma_1^i}^2) \mathrm{d}x_i, \tag{31}$$

where $\int \mathcal{N}_1(x_i; \mu_1^i, {\sigma_1^i}^2) \mathrm{d}x_i = 1$, and

$$\int (x_i - \mu_2^i)^2 \mathcal{N}_1(x_i; \mu_1^i, {\sigma_1^i}^2) \mathrm{d}x_i = \int x_i^2 \mathcal{N}_1(x_i; \mu_1^i, {\sigma_1^i}^2) \mathrm{d}x_i - 2\mu_2^i \int x_i \mathcal{N}_1(x_i; \mu_1^i, {\sigma_1^i}^2) \mathrm{d}x_i \tag{32}$$

$$+ {\mu_2^i}^2 \int \mathcal{N}_1(x_i; \mu_1^i, {\sigma_1^i}^2) \mathrm{d}x_i \tag{33}$$

$$= \mathbb{E}_{\mathcal{N}_1^i}[x^2] - 2\mu_2^i \mathbb{E}_{\mathcal{N}_1^i}[x] + {\mu_2^i}^2 \tag{34}$$

$$= (\mu_1^i - \mu_2^i)^2 + {\sigma_1^i}^2, \tag{35}$$

where $\mathcal{N}_1^i$ denotes the distribution $\mathcal{N}_1(x_i; \mu_1^i, {\sigma_1^i}^2)$. Therefore, we have

$$\int p(\boldsymbol{x}) \log q(\boldsymbol{x}) \mathrm{d}\boldsymbol{x} = -\frac{1}{2} \sum_{i=1}^{J} [\log 2\pi + \log {\sigma_2^i}^2 + \frac{(\mu_1^i - \mu_2^i)^2 + {\sigma_1^i}^2}{{\sigma_2^i}^2}]. \tag{36}$$

$\square$

**Proposition 2.** *The KL-divergence between two Gaussian distribution $p(\boldsymbol{x}) \sim \mathcal{N}_1(\boldsymbol{\mu}_1, \boldsymbol{\sigma}_1^2)$ and $q(\boldsymbol{x}) \sim \mathcal{N}_2(\boldsymbol{\mu}_2, \boldsymbol{\sigma}_2^2)$ can be calculated as:*

$$D_{\text{KL}}(p(\boldsymbol{x})||q(\boldsymbol{x})) = -\frac{1}{2} \sum_{i=1}^{d} [1 + \log(\frac{{\sigma_1^i}^2}{{\sigma_2^i}^2}) - \frac{(\mu_1^i - \mu_2^i)^2 + {\sigma_1^i}^2}{{\sigma_2^i}^2}], \tag{37}$$

*where $d$ is the dimension of parameters.*

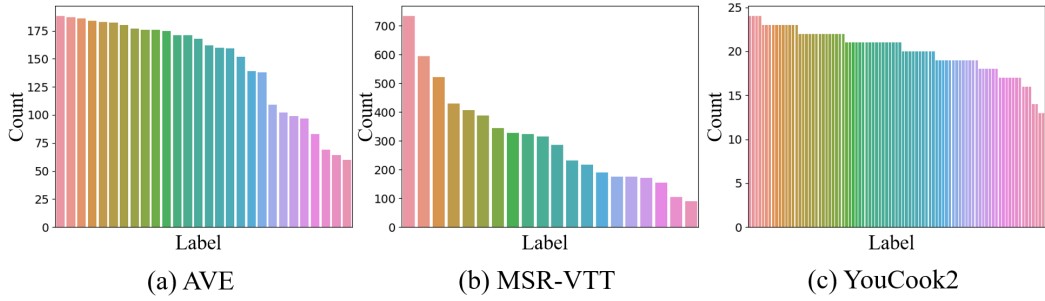

Figure 4: Distribution of Labels.

*Proof.* According to Lemma 1, we have

$$D_{\mathrm{KL}}(p(\boldsymbol{x})||q(\boldsymbol{x})) = \int p(\boldsymbol{x}) \log p(\boldsymbol{x}) \mathrm{d}\boldsymbol{x} - \int p(\boldsymbol{x}) \log q(\boldsymbol{x}) \mathrm{d}\boldsymbol{x} \tag{38}$$

$$= -\frac{1}{2} \sum_{i=1}^{d} [1 + \log(\frac{\sigma_1^{i\,2}}{\sigma_2^{i\,2}}) - \frac{(\mu_1^i - \mu_2^i)^2 + \sigma_1^{i\,2}}{\sigma_2^{i\,2}}]. \tag{39}$$

□

To specify the second part of the MFB loss, we assume the distribution $p$ and $q$ are both Gaussian. Therefore, the maximum likelihood estimation for the parameters $\boldsymbol{\mu}$ and $\boldsymbol{\sigma}^2$ of $p(\boldsymbol{Y})$ are

$$\hat{\boldsymbol{\mu}} = \frac{\sum_i^n \boldsymbol{y}_i}{n}, \hat{\boldsymbol{\sigma}}^2 = \frac{\sum_i^n (\boldsymbol{y}_i - \hat{\boldsymbol{\mu}})}{m}, \tag{40}$$

where $\boldsymbol{Y} = \{\boldsymbol{y}_1, \boldsymbol{y}_2, \ldots, \boldsymbol{y}_n\}$. Since there is no prior knowledge of the distribution $q$, we assume it to be the standard Gaussian distribution with parameters $\boldsymbol{0}$ and $\boldsymbol{I}$. Hence, the MFB loss can be calculated as:

$$\mathcal{L}_{\mathrm{MFB}} = -\sum_m \frac{1}{n} \sum_{i=1}^{n} \log \frac{\exp(\boldsymbol{x}_{m,i}^{(0)} \boldsymbol{y}_{s,i}^{(l)}/\tau)}{\sum_{j=1}^n \exp(\boldsymbol{x}_{m,i}^{(0)} \boldsymbol{y}_{s,j}^{(l)}/\tau)} - \frac{\beta}{2} \sum_{i=1}^{d} (1 + \log(\hat{\sigma}_i^2) - \hat{\mu}_i^2 - \hat{\sigma}_i^2), \tag{41}$$

where $\hat{\boldsymbol{\mu}} = \frac{\sum_i^n \boldsymbol{y}_{s,i}^{(l)}}{n}, \hat{\boldsymbol{\sigma}}^2 = \frac{\sum_i^n (\boldsymbol{y}_{s,i}^{(l)} - \hat{\boldsymbol{\mu}})}{m}$, and the first part is known as InfoNCE loss van den Oord et al. (2018) .

## D DATASET DETAILS

Further dataset details are provided in this section.

- AVE Tian et al. (2018): This dataset comprises 4,143 videos spanning 28 audio-visual event categories, which we use as ground-truth labels for clustering. The AVE dataset contains only two modalities, namely video and audio, and does not provide any text information.

- MSR-VTT Xu et al. (2016): This is a large-scale open-domain video captioning dataset consisting of 10,000 video clips across 20 categories, which we utilize as ground-truth labels for clustering. The MSR-VTT dataset presents three modalities: video, audio, and text. Each video clip is annotated with 20 English sentences, from which we randomly select one to represent the text information.

- YouCook2 Zhou et al. (2018): This is a substantial task-oriented, instructional video dataset, containing 2,000 untrimmed videos from 89 cooking recipes. We use the recipe categories as ground-truth labels for clustering. Like MSR-VTT, YouCook2 also provides three modalities: video, audio, and text. Each video's procedure steps are described in English sentences, and we randomly choose one as the text information.

The distribution of labels in each dataset is represented by a bar chart in Fig. 4. Each bar in the chart represents a label, and its height corresponds to the number of samples belonging to that label. As observed, the AVE dataset comprises 28 labels, with each label containing an average of 146.32 samples and a standard deviation of 42.31. The MSR-VTT dataset contains 20 labels, each with an average of 308.8 samples and a standard deviation of 167.82. The YouCook2 dataset has 89 labels, each averaging 20.11 samples with a standard deviation of 2.51. Hence, the label distribution in the MSR-VTT dataset is highly uneven, while YouCook2's distribution is relatively balanced, and AVE's distribution lies somewhere in between. The unevenness in label distribution could pose a challenge to our model's learning process due to the imbalanced representation across different classes. Nonetheless, as seen in the experimental results, our method outperforms other techniques across all datasets, indicating its robustness against imbalances in labels and suggesting strong generalization capabilities of our model.

## E    IMPLEMENTATION DETAILS

We extract features following the methodology described in MCN Chen et al. (2021). Specifically, for video features, we leverage a combination of pre-trained 2D features from a ResNet152 model He et al. (2016) and pre-trained 3D features from a ResNeXt-101 model Hara et al. (2018). Audio features are extracted using log-mel spectrograms and a pre-trained DAVEnet model Harwath et al. (2020). In the textual branch, sentence embeddings are created by applying max-pooling to word embeddings, which are generated using a GoogleNews pre-trained Word2vec model Mikolov et al. (2013). Throughout training, all these backbone components are kept fixed.

To manage the complexity of the multimodal data, we employ an auto-encoder to reduce the dimensionality to 256. This auto-encoder consists of one or two layers, each of which includes a linear layer, a batch normalization layer, a ReLU activation layer, and a dropout layer with a rate of 0.5. The optimization of the auto-encoder is done through mean squared error (MSE) reconstruction loss.

The hyperparameters of our model are set as follows: For the optimization process, we use the Adam optimizer Kingma & Ba (2015) with a learning rate of $1 \times 10^{-4}$ and a weight decay $1 \times 10^{-3}$. We also use a step learning rate scheduler every 20 steps with a rate of 0.5. In the construction of the hypergraph, the $k$ value for the KNN method is set as 7, and the number of hypergraph layers $L$ is set as 2. Lastly, the hyper-parameter $\beta$ is set as 0.2. The sensitivity analysis of hyperparameters can be found in the appendix. We leverage the K-means algorithm for clustering, using the pre-set number of clusters as defined in each dataset. All experiments are conducted on a server with two Intel Xeon E5-2678 2.50 GHz CPUs and an Nvidia GeForce RTX 3090 GPU.

## F    COMPUTATION OF METRICS

Metrics for clusteirng task are calculated as follows. Accuracy (Acc) measures agreement between true labels $\boldsymbol{y}_i$ and clustering labels $\hat{\boldsymbol{y}}_i$, given by

$$Acc = \frac{\sum_{i=1}^{n} \delta(\boldsymbol{y}_i, \hat{\boldsymbol{y}}_i)}{n}, \tag{42}$$

where $n$ is the total number of samples. Normalized Mutual Information (NMI) quantifies the shared information, expressed as

$$NMI = \frac{2 \cdot I(\boldsymbol{y}; \hat{\boldsymbol{y}})}{H(\boldsymbol{y}) + H(\hat{\boldsymbol{y}})}, \tag{43}$$

with $I$ representing mutual information and $H$ representing entropy. Adjusted Rand Index (ARI) measures similarity corrected for chance, given by

$$ARI = \frac{RI - E[RI]}{\max(RI) - E[RI]}, \tag{44}$$

where $RI$ is the Rand Index and $E[RI]$ is its expected value under random assignment.

Table 5: Experiment results of further ablation studies.

| Dataset | AVE | | | MSR-VTT | | | YouCook2 | | |
|---|---|---|---|---|---|---|---|---|---|
| Ablations | **Acc** | **NMI** | **ARI** | **Acc** | **NMI** | **ARI** | **Acc** | **NMI** | **ARI** |
| w/o $\mathcal{E}_m$ | $66.8 \pm 1.6$ | $74.9 \pm 0.7$ | $59.2 \pm 1.2$ | $40.6 \pm 1.2$ | $36.7 \pm 0.2$ | $26.9 \pm 1.8$ | $29.4 \pm 0.4$ | $55.9 \pm 0.5$ | $15.4 \pm 0.5$ |
| w/o attention | $67.7 \pm 1.8$ | $74.6 \pm 0.9$ | $58.7 \pm 1.4$ | $38.7 \pm 1.3$ | $36.3 \pm 0.2$ | $24.0 \pm 1.4$ | $29.4 \pm 0.7$ | $55.8 \pm 0.5$ | $15.5 \pm 0.8$ |
| full model | $68.3 \pm 2.3$ | $75.7 \pm 1.1$ | $60.7 \pm 2.0$ | $41.8 \pm 0.5$ | $37.0 \pm 0.3$ | $28.8 \pm 1.0$ | $29.6 \pm 1.1$ | $56.9 \pm 0.9$ | $16.3 \pm 1.0$ |

## G  DETAILS OF TEXT-TO-VIDEO RETRIEVAL TASK AND TEMPORAL ACTION LOCALIZATION TASK

In our methodology, both tasks are implemented in a semi-supervised manner. We construct the hypergraph structure utilizing both the training and testing datasets, adhering to the original dataset splits Xu et al. (2016); Zhukov et al. (2019).

For the text-to-video retrieval task, within the testing set's instance hypergraph, we restricted relationships to the video and audio modalities only, deliberately excluding links between text and video. This exclusion is due to the inherent uncertainty of relationships between text and instances in this task.

For the temporal action localization task, the textual data provides step information. While a single task may consist of numerous clips, it often contains only a handful of steps. Similar to the text-to-video retrieval task, the $\boldsymbol{H}_s^t$ is constructed solely from the training set. It's worth noting that, in this setting, instances in the testing set do not have textual information. This demonstrates our method's adaptability even in scenarios with missing modalities.

## H  ADDITIONAL ABLATION STUDIES

In this section, we conduct additional ablation studies, as shown in Table 5.

**Ablation study of modality hyperedge** $\mathbb{E}_m$**.** In previous ablation studies, we replaced the modality hypergraph $\boldsymbol{H}_m$ with an identity matrix to convert the hypergraph propagation layer into a linear layer. This demonstrates the significance of high-order correlations. Nevertheless, we are curious about the impact of removing the entire modality hyperedge $\mathbb{E}_m$, as it doesn't participate in the loss function (Eq. 41). Therefore, the hypergraph propagation process converts into:

$$\boldsymbol{Y}_s^{(0)} = \frac{1}{3}(\boldsymbol{X}_v^{(0)} + \boldsymbol{X}_a^{(0)} + \boldsymbol{X}_t^{(0)}), \tag{45}$$

$$\boldsymbol{Y}_s^{(l+1)} = f^{attn}(\boldsymbol{X}^{(l)}, \boldsymbol{Y}_s^{(l)}, \boldsymbol{H}_s^{\top}), \tag{46}$$

$$\boldsymbol{X}^{(l+1)} = f^{attn}(\boldsymbol{Y}_s^{(l+1)}, \boldsymbol{X}^{(l)}, \boldsymbol{H}_s). \tag{47}$$

This implies that we disregard correlations within the same modality, focusing instead on cross-modality high-order correlations within the same instance. Moreover, this indicates that the vertex information of the $l+1$-th layer $\boldsymbol{X}^{(l+1)}$ comes solely from the instance hyperedge, which could lead to an oversmoothing problem.

As depicted in Table 5, we observe that the full model outperforms the version without the modality hyperedge $\mathbb{E}_m$ on average by 1.90%, 3.61%, and 2.77% for the AVE, MSR-VTT, and YouCook2 datasets, respectively. As we mentioned above, our method uses pre-trained features, which already consider correlations within the same modality during the pre-training process. Thus, even when intra-modality correlations are not considered, competitive performance can still be achieved. Notably, when we further consider high-order intra-modality correlations, the performance improves as it not only considers high-order correlations within the same modality but also prevents the oversmoothing problem. Therefore, the effectiveness of modality hyperedge $\mathbb{E}_m$ and modality hypergraph $\boldsymbol{H}_m$ is demonstrated.

Table 6: Experimental results on computational complexity. The training and testing time are presented for 100 epochs, excluding the time taken for metric computation.

| Dataset | Model | train time | test time | GFLOPs | parameters |
|---------|-------|-----------|-----------|--------|------------|
| AVE | AGE | 5.28 | 0.04 | 10.49 | 2,560,500 |
| | AHAGE | 0.59 | 0.06 | 10.49 | 2,560,500 |
| | MCN | 1299.95 | 562.03 | 3177.06 | 187,805,954 |
| | MFLVC | 46.68 | 15.04 | 3.68 | 14,314,172 |
| | HyperRep | 3.21 | 0.55 | 54.41 | 12,745,728 |
| MSR-VTT | AGE | 21.87 | 0.04 | 34.78 | 5,632,500 |
| | AHAGE | 1.46 | 0.06 | 34.78 | 5,632,500 |
| | MCN | 1593.84 | 995.66 | 3219.64 | 265,165,058 |
| | MFLVC | 76.13 | 24.23 | 3.68 | 14,310,068 |
| | HyperRep | 17.97 | 0.74 | 167.64 | 26,206,720 |
| YouCook2 | AGE | 1.98 | 0.04 | 10.08 | 5,632,500 |
| | AHAGE | 0.30 | 0.10 | 10.08 | 5,632,500 |
| | MCN | 1056.32 | 907.86 | 3219.64 | 265,165,058 |
| | MFLVC | 18.33 | 7.42 | 3.70 | 14,345,465 |
| | HyperRep | 3.63 | 0.89 | 14.76 | 7,313,920 |

**Ablation study of attention mechanism.** We further conduct an ablation experiment on the attention mechanism. The hypergraph propagation process becomes:

$$\boldsymbol{Y}_s^{(l+1)} = f^p(\boldsymbol{X}^{(l)}, \boldsymbol{H}_s^\top), \tag{48}$$

$$\boldsymbol{Y}_m^{(l+1)} = f^p(\boldsymbol{X}^{(l)}, \boldsymbol{H}_m^\top), \tag{49}$$

$$\boldsymbol{X}^{(l+1)} = f^p([\boldsymbol{Y}_s^{(l+1)} \| \boldsymbol{Y}_m^{(l+1)}], [\boldsymbol{H}_s \| \boldsymbol{H}_m]). \tag{50}$$

This indicates that we treat each vertex and each hyperedge with equal attention.

As shown in Table 5, we observe that the full model outperforms the version without attention mechanism by an average of 1.92%, 9.98%, and 2.60% on the AVE, MSR-VTT, and YouCook2 datasets, respectively. This suggests that the attention mechanism in the hypergraph allows the model to assign different levels of attention to information from different vertices or hyperedges, thereby enhancing performance.

## I COMPUTATIONAL COMPLEXITY ANALYSIS

To demonstrate the practical applicability of the proposed method, We present computational complexity analysis. We first conduct a qualitative analysis of the computational complexity. For a context with $n$ instances, $m$ modalities, a feature dimension of $d$, and $k$ as the hyperparameter for K-NN hypergraph construction, the computational complexity for hypergraph construction amounts to $O(mn^2)$ and $O(dmn^2 + mnklog(n) + mnk)$ for the instance hypergraph $\boldsymbol{H}_s$ and modality hypergraph $\boldsymbol{H}_m$, respectively. Next, we turn our attention to the Hypergraph Propagation Module which are described in Eq.8 to Eq.11. Respectively, these equations bring computational complexities of $O(nd)$, $O(nd^2 + mnd^2 + n^2md + mn)$, $O(mnkd + mnd^2)$, and $O((m+2)nd^2 + (m+1)mn^2d^2 + (k+1)mn)$. Given that both $m$ and $k$ are significantly small compared to $n$ and $d$, the computational complexity of both the hypergraph construction module and hypergraph propagation module can be summarized as $O(n^2d)$ and $O(nd^2 + n^2d)$, respectively. It's important to note that the hypergraph construction process is not an inherent component of our model, but rather a preprocessing step for the data. Nonetheless, we have included its computational complexity analysis to provide reviewers with a thorough understanding of our entire methodology.

Second, we present the results of our analysis experiments concerning the complexity of the proposed method in Table 6. As evident, our method boasts relatively low training and testing time, accompanied by reasonable GFLOPs and parameters. Such efficiency in training and testing time underscores the scalability of our method, making it suitable for larger datasets and real-world deployment scenarios. The optimal balance between GFLOPs and parameters further indicates that our method is computationally efficient, without compromising the model's capacity. This is crucial for practical applications, especially in environments with limited computational resources. Moreover, having a lower computational footprint while maintaining superior performance, as observed in previous results, is a testament to the method's effectiveness and efficiency. It highlights that our approach

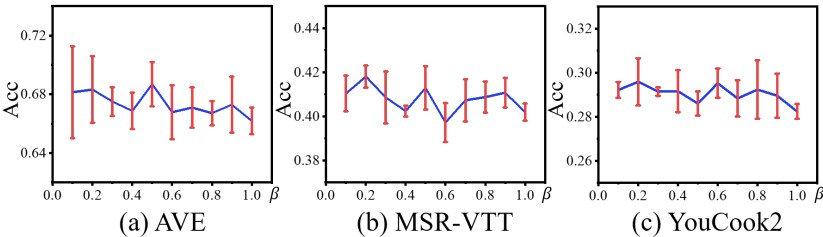

Figure 5: Accuracy variations when altering the value of $\beta$ in MFB loss (Eq. 14) across three datasets.

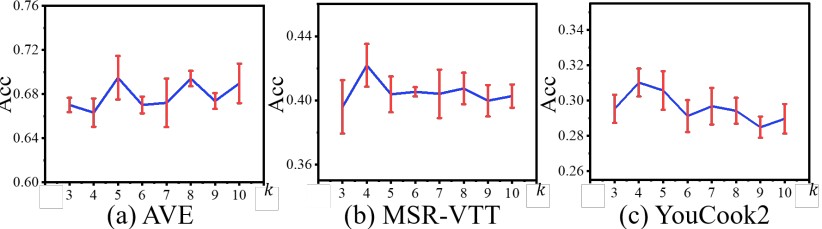

Figure 6: Accuracy variations when altering the value of $k$ in $k$-NN for hypergraph construction across three datasets.

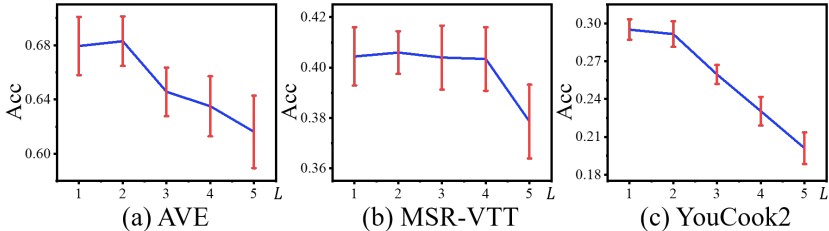

Figure 7: Accuracy variations when altering the value of the number of hypergraph layer $L$ across three datasets.

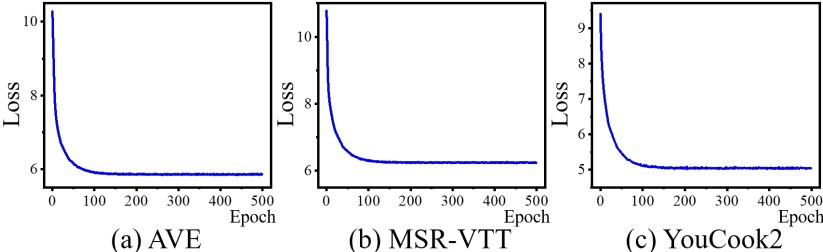

Figure 8: Curves of training loss on three datasets.

doesn't just rely on model size or computational might but on innovative techniques and strategies that ensure meaningful multi-modal representation learning.

## J    SENSITIVITY ANALYSIS AND CONVERGENCE ANALYSIS

To investigate the robustness of our model and identify key influencing hyper-parameters, we conducted sensitivity analyses. Firstly, we varied the value of $\beta$ from 0.1 to 1.0 as per Eq.14, the results of which are displayed in Fig.5. As depicted, the accuracy remains relatively stable across different values of $\beta$, albeit with a slight decreasing trend as $\beta$ increases. This suggests that our model is largely insensitive to $\beta$. However, as $\beta$ increases and the constraint tightens, there is a gradual effect on the performance of the model.

Next, we varied the value of $k$ from 3 to 10 for the $k$-NN algorithm used for constructing hypergraphs. The results, shown in Fig. 6, generally demonstrate that the model's performance isn't dramatically affected by different values of $k$. However, a slight performance decrease is observed with increasing $k$ on the YouCook2 dataset. This could be attributed to YouCook2 being a relatively smaller dataset, where the use of larger hyperedges may introduce noise. Regardless, these findings suggest that, for most cases, fine-tuning this specific hyper-parameter when constructing hypergraphs may not be strictly necessary.

Moreover, as illustrated in Fig 7, the model exhibits strong performance across all datasets when $L$ is set to 1 or 2. However, as $L$ increases, a noticeable decline in performance is observed. This trend can be attributed to the well-known over-smoothing problem, where all vertex features tend to converge and become indistinguishable in the feature space. As a result of this issue, hypergraph neural networks typically avoid deep architectures, and it is common practice to select a value of 2 for layer number $L$.

Lastly, we analyze the convergence of HyperRep by tracking the value of the loss function 14 during training over 500 epochs across three datasets. As shown in Fig. 8, the model exhibits a steady decrease in loss, indicating effectively learning from the data. The model reaches a stable state after approximately 200 epochs, suggesting efficient convergence. This rapid convergence is beneficial in practical applications, reducing the time and computational resources required for model training.

## K  QUALITATIVE ANALYSIS

As shown in Fig. 9, we provide qualitative results of the text-to-video retrieval task on the MSR-VTT dataset of HyperRep, MCN, and MIL-NCE. Given a specific text, our model presents the top 5 videos it recalls as being most relevant. The videos encircled in red represent the ground truth matches. From this visualization, it's evident that our model adeptly captures the semantic nuances embedded within the text modality and successfully maps them to the corresponding segments in the video modality. The consistency between the textual description and the retrieved videos shows the model's ability to effectively combine information from different modalities. Such precision not only showcases the robustness of our model's architecture but also its ability to discern intricate semantic relationships. The multimodal representations learned by our approach bridge the semantic gap between text and video, making it a powerful tool for tasks that require deep understanding across modalities.

## L  INTERPRETABLE ANALYSIS

As shown in Fig. 10, the distinct clusters formed by the data points illustrate the efficacy of our model in learning separable and interpretable representations. Each color in the visualization represents a different category, revealing how well the HyperRep framework groups instances with high semantic similarity.

The visualization highlights that representations learned with MFB loss (Fig.10 (d)) result in more distinct and cohesive clusters compared to those relying solely on InfoNCE (Fig.10 (c)). This supports the quantitative findings that high-order correlations play a significant role in the final performance scores, demonstrating their importance in capturing complex multimodal interactions that InfoNCE alone might not fully encapsulate.

Moreover, the ablation study without high-order correlation (Fig.10 (b)) falls short in terms of clustering quality, as evidenced by the more dispersed clusters and less distinct group boundaries. This visually corroborates the quantitative analysis, which shows a notable drop in performance metrics when high-order correlations are omitted, underscoring their role in enhancing the discriminative power of the learned representations.

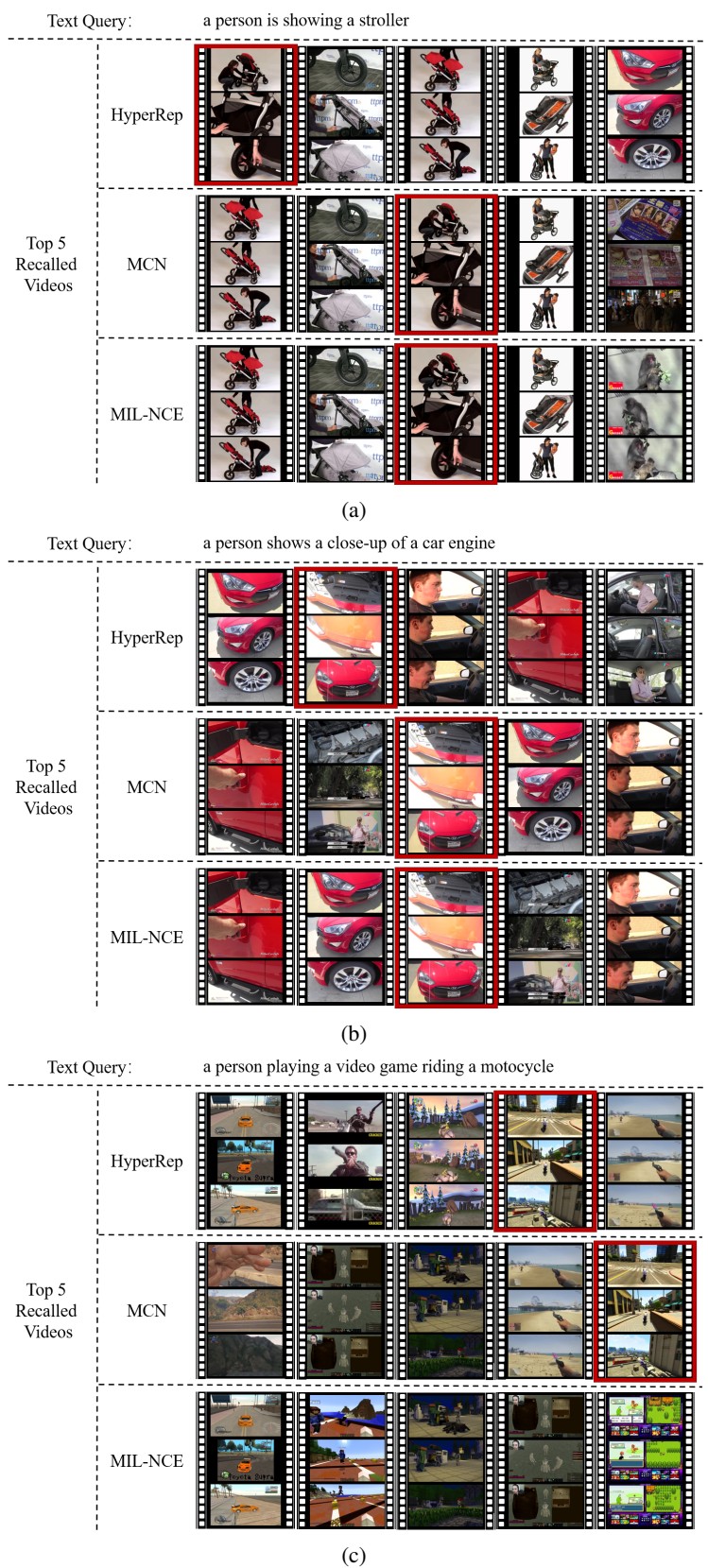

Figure 9: Qualitative results for the text-to-video retrieval task on MSR-VTT dataset.

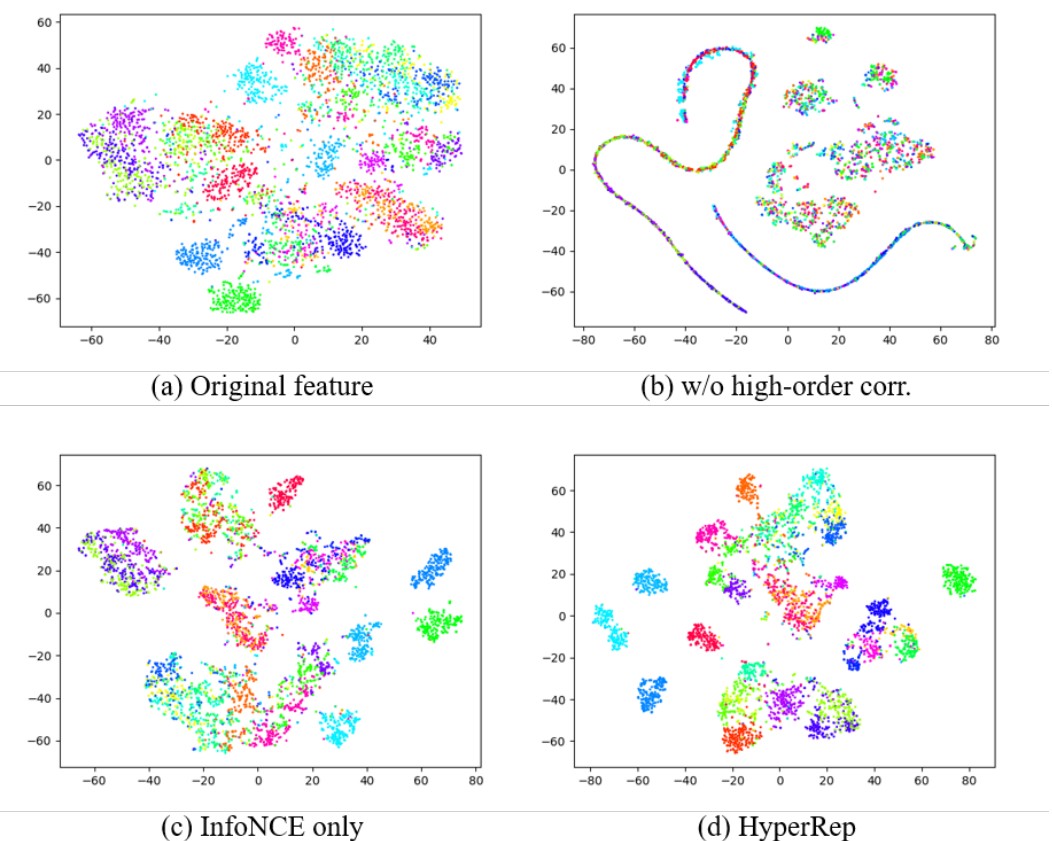

Figure 10: The t-SNE visualization of multimodal data representations on AVE dataset, with each color corresponding to a different data category/class.

