# OpenReview forum: "HyperRep: Hypergraph-Based Self-Supervised Multimodal Representation Learning"
_ICLR.cc/2024/Conference — Submitted to ICLR 2024_

### Official Review · Reviewer_N99Z · 2023-10-31

**Soundness:** 3 good
**Presentation:** 3 good
**Contribution:** 3 good
**Rating:** 8
**Confidence:** 2

**Summary:**

This paper introduces HyperRep, a hypergraph-based self-supervised multimodal representation learning method that captures high-order correlations in real-world multimodal data. The proposed approach balances the benefits of contrastive methods and preserves the unique aspects of each data point, achieved through the construction of dual types of hypergraphs. The paper presents experimental results on several downstream tasks, demonstrating that HyperRep delivers consistently competitive results against state-of-the-art methods.

**Strengths:**

The idea of using hyperedge to facilitate multi-modal fusion is interesting.

The paper is clear and easy to follow.

**Weaknesses:**

For the Hypergraph propagation module figure (Fig. 3), a more detailed and clearer introduction should be helpful.

There are a few typos. Please find and correct them.

**Questions:**

1. May the author explain whether  HyperRep requires pre-trained encoder for each modal or it trains these encoders using the MFB loss? If pre-trained encoders are used, may the author provide ablation about the encoders used in the experiment?
2. May the author explain the intuition behind using the instance hyperedge features for the downstream task, not the instance feature itself?

---

> ### Author Response · Authors · 2023-11-17
> **Response to Reviewer N99Z**
>
> We thank the reviewer for the careful reading and valuable feedback on our submission. We are deeply encouraged that the reviewer appreciated our idea and the presentation is clear and we apologize for the typos and have corrected them in the revised manuscript. The acceptance of our work is particularly heartening, and we extend our sincere thanks for the insightful review. Here we provide more explaination towards detailed descriptions of our method. We hope that our response addresses your concerns.
>
> **[Concern 1]** On the introduction on hypergraph propagation module.
>
> Thank you for your feedback regarding the clarity of Figure 3, which illustrates our Hypergraph propagation module.
>
> In essence, the module applies an attention mechanism to the vertices, which represent features from each modality, and the hyperedges, which aggregate these features to capture instance-level semantics. Attention weights are determined via a scaled dot-product attention function, allowing the model to dynamically emphasize the most relevant modal features during aggregation. The process involves iteratively updating hyperedge features by aggregating information from the vertices, followed by updating the vertices with the new hyperedge information. This bi-directional flow ensures that both modality-specific and cross-modal information is captured and refined over multiple iterations. The final output after \\(L\\) layers of propagation is a set of hyperedge features that embody the learned instance representation, ready for downstream tasks. The whole process can be succinctly described by Eq.8-11 in our manuscript.
>
> **[Concern 2]** On the pre-trained encoder issue.
>
> Thank you for highlighting the encoder aspect of our method. As detailed in Appendix F of our manuscript, HyperRep utilizes pre-trained models for feature extraction from different modalities: ResNet for video, DAVEnet for audio, and Word2vec for textual modality features​​. The choice of these pre-trained models is motivated by their proven effectiveness in capturing rich, modality-specific features, which are essential for multimodal tasks.
>
> Regarding the concern about training from scratch, it's widely acknowledged in the multimodal learning community that most methods benefit from pre-extracted features or a fixed backbone due to challenges in aligning different modalities and the prohibitive computational costs associated with training complex models from scratch. HyperRep focuses on leveraging initial multimodal features to learn instance representations, rather than retraining these backbones. Therefore, ablating the pre-trained models is not feasible within the scope of our approach.
>
> In terms of the experiments, we have ensured a fair comparison by using identical input features across methodologies wherever possible. For instance, in the clustering task, except for SeLaVi, which operates directly on raw videos, all other compared methods use the same input features as HyperRep. This consistent approach extends to our text-to-video retrieval and action localization tasks, where several methods employ the same encoders as ours. We also include methods using different backbones for a comprehensive comparison, as shown in Table 2 and Table 4.
>
> **[Concern 3]** On the instance feature utilzation.
>
> Thank you for the opportunity to clarify the terminology used in our manuscript. We would like to emphasize that in the context of our HyperRep model, the terms 'instance feature' and 'instance hyperedge feature' are used interchangeably and denote the same concept. The apparent distinction is a matter of terminology, for which we apologize for any confusion.
>
> To elucidate further, our model does not utilize raw instance features directly from the input data. Instead, it learns a composite representation that encapsulates the combined information of multiple modalities, which we refer to as the 'instance hyperedge feature'. This learning is achieved through a hypergraph propagation process where individual vertex features (modality features) are aggregated to form hyperedges, representing the instances. Thus, the 'instance hyperedge feature' effectively serves as the learned 'instance feature'.

---

### Official Review · Reviewer_scVn · 2023-10-31

**Soundness:** 3 good
**Presentation:** 2 fair
**Contribution:** 3 good
**Rating:** 6
**Confidence:** 3

**Summary:**

The paper proposes HyperRep, a self-supervised representation learning method for multimodal data that leverages hypergraphs to capture high-order inter- and intra-modality correlations. HyperRep also uses an information bottleneck principle to fuse multimodal data effectively. The paper shows that HyperRep outperforms existing methods on various downstream tasks.

**Strengths:**

1. This paper provides sufficient and detailed formal definitions and necessary proofs.
2. The application of hypergraphs in this paper may inspire the field of multimodal self-supervised learning.
3. The authors conducted extensive ablation experiments on multiple datasets.

**Weaknesses:**

1. Motivation: The paper highlights the use of hypergraphs to capture higher-order correlations among modalities. However, I find the paper insufficient in explaining and analyzing why higher-order correlations are essential for multimodal self-supervised learning. This is especially relevant since most existing methods, such as the CLIP[1] series and ImageBind[2], rely on pair-wise multimodal self-supervision.
2. Method: The paper could improve the clarity and presentation of MFB. For instance, how does Eq. 12 relate the mutual information of shared and instance features? A clearer explanation of the illustration would be helpful. Also, the MFB module in Figure 2 is vague, and a more specific illustration could enhance the readers’ comprehension of the method.
3. Experiment: The paper lacks comparisons with recent (2023) methods, as the only one mentioned has a huge performance gap with the classical methods. Could the paper also compare with more popular Foundation models, such as VideoCLIP[3] and OmniVL[4]? Moreover, Table 1 shows that graph-based methods perform poorly, and Table 2 reveals a drastic performance drop after removing higher-order correlations. However, AGC achieves high performance without using higher-order correlations. Does this imply that higher-order correlations are crucial for the proposed method, but not for the multimodal self-supervised learning task?

[1] Learning Transferable Visual Models From Natural Language Supervision. ICML 2021.
[2] IMAGEBIND: One Embedding Space To Bind Them All. ICCV 2023.
[3] VideoCLIP: Contrastive Pre-training for Zero-shot Video-Text Understanding. EMNLP 2021.
[4] OmniVL: One Foundation Model for Image-Language and Video-Language Tasks. NeurIPS 2022.

**Questions:**

How come the MIL-NCE performance in Table II differs so much from the original paper? Are there any differences in the experimental settings?

---

> ### Author Response · Authors · 2023-11-17
> **Response to Reviewer scVn (1)**
>
> We thank the reviewer for the careful reading and valuable feedback on our submission. We are deeply encouraged by the reviewer's appreciation of our work's interesting ideas and the detailed, extensive nature of our experiments. In response, we provide more detailed explanations and descriptions to further clarify the motivation behind our method and to present additional experimental results. We hope that our response comprehensively addresses your concerns. We look forward to any further feedback or suggestions you might have.
>
> **[Concern 1]** On motivation of high-order correlations.
>
> Thank you for your insightful query regarding the motivation for integrating high-order correlations in multimodal self-supervised learning. As elucidated in our manuscript, high-order correlations are a natural extension and generalization of pairwise correlations, effectively encompassing and expanding upon them.
>
> In the realm of hypergraphs, which we utilize in HyperRep, a conventional graph (encompassing pairwise correlations) is essentially a 2-uniform hypergraph where each edge connects exactly two vertices. By extending this concept to high-order correlations, we are not only capturing pairwise relationships but also encompassing more complex interactions that involve multiple modalities simultaneously. This is crucial for a thorough and nuanced understanding of multimodal data, which often involves intricate relationships that go beyond simple pairwise interactions.
>
> While conventional methods focusing on pairwise supervision have demonstrated effectiveness, they are often insufficient for modeling the complexity inherent in multimodal datasets. These complexities are often inherent in multimodal scenarios but are underrepresented in pairwise models. For example, as we illustrate in Fig.1(a) of our paper, a multifaceted scenario like a video of a car drifting involves complex correlations with related images, engine sounds, and textual descriptions that go beyond simple pairwise associations.
>
> High-order correlations naturally encapsulate the pairwise interactions while providing a broader and more comprehensive framework for understanding the complex dynamics of multimodal interactions. This is why our HyperRep approach leverages hypergraphs to model these correlations, thus capturing a richer and more intricate representation of multimodal data than would be possible with a focus solely on pairwise relationships. It sets HyperRep apart from existing methods and underscores the unique value it brings to the field.
>
> Therefore, our focus on high-order correlations is rooted in their ability to encompass and go beyond pairwise relationships, offering a more complete and effective approach to multimodal representation learning.
>
> **[Concern 2]** On clearer presentation.
>
> Thank you for your valuable feedback on improving the presentation and clarity of the Multimodal Fusion Bottleneck (MFB) in our work.
> In Figure 2, the MFB module is depicted as encompassing the principles represented in Equations 12 and 13. The MFB principle is fundamentally about balancing the mutual information between shared and instance features, as elaborated in the introduction section in the manuscript. Specifically, it aims to maximize the mutual information between each modality and the instance, while simultaneously minimizing the mutual information between the instance and the aggregated modality information. This principle is visually represented in Fig. 1(c), where the white and shaded circles denote modality and instance information, respectively, and the slashed area symbolizes the cross-modality shared information. The goal of MFB is to ensure that the instance information (represented by the shaded circle) covers as much of the shared multimodal information (denoted by the slashed area) as possible. This alignment effectively constrains the solution space, directing the model’s learning process to focus primarily on the shared information across modalities.

---

> ### Author Response · Authors · 2023-11-17
> **Response to Reviewer scVn (2)**
>
> **[Concern 3]** On comparison with recent and popular methods.
>
> We are grateful for your suggestion to broaden our comparison with more recent methods. While OmniVL [2] does not have available implementation code for direct comparison within our current time constraints, we have included additional comparisons with popular and recent methods [3,4,5] from 2022 and 2023 in following charts. The best and second-best performances are highlighted in **bold** and *italic* respectively.
>
> From the following results, it is evident that HyperRep excels in the text-to-video retrieval task on the MSR-VTT dataset, achieving the best performance across all metrics. On the action localization task on the CrossTask dataset, HyperRep attains the best performance in Recall. This demonstrates the model's effectiveness against recent and popular methods.
>
> These results affirm the state-of-the-art performance of HyperRep and its significant contributions to the field of multimodal learning, illustrating the model's robustness and the impact of our novel approach even when compared with the latest methodologies.
>
> **Text-to-video retrieval task on MSR-VTT dataset**:
> | Method | Modality | Model | TR | R@1 | R@5 | R@10 |
> |:-------------|:--------------:|:--------------:|:-------------:|:--------------:|:--------------:|:--------------:|
> | VideoCLIP [1] | VT | S3D | N | *10.4* | 22.2 | 30.0 |
> | VT-TWINS [3] | VT | S3D | N | 9.4 | 23.4 | 31.6 |
> | Shvetsova et al. [4] | VAT | R152 + RX101 | Y | 10.3 | *24.6* | *35.3* |
> | Shvetsova et al. [4] | VAT | S3D | Y | 9.9 | 24.0 | 32.6 |
> | HyperRep | VAT | R152 + RX101 | N | **11.6** | **26.3** | **37.3** |
>
> **Action localization task on CrossTask dataset**:
> | Method | Modality | Model | TR | Recall |
> |:-------------|:--------------:|:--------------:|:-------------:|:--------------:|
> | VideoCLIP [1] | VT | S3D | N | 33.9 |
> | VT-TWINS [3] | VT | S3D | N | 40.7 |
> | Shvetsova et al. [4] | VAT | R152 + RX101 | N | 39.3 |
> | Shvetsova et al. [4] | VAT | S3D | N | 41.1 |
> | Simsek et al. [5] | VT | S3D | N | *41.76* |
> | HyperRep | VAT | R152 + RX101 | N | **50.68** |
>
> [1] VideoCLIP: Contrastive Pre-training for Zero-shot Video-Text Understanding. EMNLP 2021.
> [2] OmniVL: One Foundation Model for Image-Language and Video-Language Tasks. NeurIPS 2022.
> [3] Video-Text Representation Learning via Differentiable Weak Temporal Alignment. CVPR 2022.
> [4] Everything at Once -- Multi-modal Fusion Transformer for Video Retrieval. CVPR 2022.
> [5] Learning actionness from action/background discrimination. Signal Image Video Process 2023.

---

> ### Author Response · Authors · 2023-11-17
> **Response to Reviewer scVn (3)**
>
> **[Concern 4]** On high-order correlation issue.
>
> Thank you for your insightful question regarding the role of high-order correlations in multimodal self-supervised learning. As indicated in our manuscript, high-order correlations are essential for comprehensively understanding multimodal data. They encapsulate intricate relationships within and across modalities that simple pairwise correlations might not fully capture. These complex relationships are crucial for thoroughly understanding the data and are often overlooked by existing methods that predominantly focus on pairwise interactions.
>
> While AGC, a graph clustering method, achieves commendable performance without explicitly using high-order correlations, it's important to note that the graph structure it employs can inherently contain high-order information. In our experiments, we construct a graph adjacency matrix $\boldsymbol{A}=\boldsymbol{HH^\top}$ using the hypergraph incidence matrix $\boldsymbol{H}$ for a fair comparison with AGC. This clique expansion approach effectively retains the high-order information originally present in the hypergraph structure.
>
> Our ablation study results reinforce this point. By setting $k=2$ for hypergraph construction, thereby reducing them to graphs, we noticed a performance decline compared to including high-order correlations ($k=7$). Specifically, HyperRep ($k=2$) underperformed compared to HyperRep ($k=7$), highlighting the importance of high-order correlations in capturing multimodal interactions' complexity. Moreover, HyperRep ($k=2$) still consistently and significantly outperforms AGC across three datasets, demonstrating our method's effectiveness.
>
> Thus, while AGC and other methods might achieve good performance without explicitly modeling high-order correlations, our findings suggest that these correlations are crucial in advancing multimodal representation learning, especially for tasks involving complex data relationships. Your question has inspired us to refine our ablation studies further, and we deeply appreciate it.
>
> AVE dataset:
> | Method | ACC | NMI | ARI |
> |:-------------|:--------------:|:--------------:|:-------------:|
> | AGC | 63.1 ± 0.4 | 70.8 ± 0.1 | 52.0 ± 0.4 |
> | HyperRep (k=2) | 65.6 ± 1.8 | 74.6 ± 0.6 | 58.0 ± 1.0 |
> | HyperRep (k=7) | 68.3 ± 2.3 | 75.7 ± 1.1 | 60.7 ± 2.0 |
>
> MSR-VTT dataset:
> | Method | ACC | NMI | ARI |
> |:-------------|:--------------:|:--------------:|:-------------:|
> | AGC | 36.4 ± 0.5 | 33.1 ± 0.1 | 16.9 ± 0.5 |
> | HyperRep (k=2) | 38.8 ± 1.8 | 36.6 ± 0.1 | 23.8 ± 2.5 |
> | HyperRep (k=7) | 41.8 ± 0.5 | 37.0 ± 0.3 | 28.8 ± 1.0 |
>
> YouCook2 dataset:
> | Method | ACC | NMI | ARI |
> |:-------------|:--------------:|:--------------:|:-------------:|
> | AGC | 20.5 ± 0.6 | 46.8 ± 0.6 | 6.9 ± 0.5 |
> | HyperRep (k=2) | 28.5 ± 1.2 | 55.7 ± 0.9 | 14.9 ±  1.0|
> | HyperRep (k=7) | 29.6 ± 1.1 | 56.9 ± 0.9 | 16.3 ± 1.0 |
>
> **[Concern 5]** On the performance of MIL-NCE.
>
> The performance of MIL-NCE on the text-to-video retrieval task on the MSR-VTT dataset is the same as the original paper, as showcased in Fig.5 (b), page 8 in the original paper. However, we did make a typographical error in reporting the R@5 for I3D-based MIL-NCE as 22.0; it should indeed be 22.2 as stated in the original paper. We appreciate your attention to detail and thank you for bringing this to our notice.

---

> ### Author Response · Authors · 2023-11-22
> **Gentle Reminder on Further Discussions**
>
> Dear Reviewer scVn,
>
> We express our gratitude for your valuable time in reviewing our submission. We welcome any further questions you may have regarding our work or rebuttal. Please do not hesitate to share any additional concerns. We are always open to further discussion.
>
> Sincerely,
> Authors.

---

> > ### Comment · Reviewer_scVn · 2023-11-22
> > **Thanks for the replies!**
> >
> > Thanks for the author's reply, and they mostly addressed my concerns. I've raised my score to 6.

---

> > > ### Author Response · Authors · 2023-11-22
> > > **Gratitude for Revised Review and Score Update**
> > >
> > > Dear Reviewer scVn,
> > >
> > > We sincerely appreciate your reconsideration and are heartened to hear that our responses have addressed your concerns. Your revised score is greatly encouraging. Thank you once again for your thoughtful feedback and support of our work.
> > >
> > > Warm regards,
> > > Authors

---

### Official Review · Reviewer_bsih · 2023-10-31

**Soundness:** 3 good
**Presentation:** 3 good
**Contribution:** 3 good
**Rating:** 6
**Confidence:** 4

**Summary:**

This paper aims to solve the existing challenges in self-supervised representation learning, where most of the existing work overlooks the high-order inter- and intra-modality correlations characteristics and lacks effective fusion principles. To tackle these issues, the author proposed HyperRep, which combined hypergraph-based modeling and self-supervised multimodal fusion principle to achieve superior representation learning.

**Strengths:**

1. The paper is well-written and nicely-structured.

2. The paper proposed hypergraph-based representation learning creatively uses graph structure to represent the inter- and intra-modal relationships for multi-modal data, which helps the model capture high-order correlations of the instances.

3. The proposed MFB loss leverages the bottleneck principle to encourage the model to capture the most informative aspects of the data and is demonstrated over multiple downstream tasks.

4. The authors conducted diverse ablation studies that demonstrate the effectiveness of different components of the proposed method and the ability of handling missing modalities. This makes the method suitable for real-world applications.

5. There is a consistent improvement in performance over all three downstream tasks, clustering, text to video retrieval and action localization over prior baselines.

**Weaknesses:**

1. In the hypergraph construction process, which is a preprocessing step for the data, has a computational complexity of O(n^2d). This may limit the scalability of the method for datasets with large amount of instance or more complex data (high-resolution images).

2. Lacking experiment that provides a detailed analysis of the interpretability of the learned representations.

3. It would be better to add the results from baseline methods in Figure 9.

4. In Table 3, it looks like row 1 (high order correlation) has a big say in the final scores, while InfoNCE doesn't seem to make much of a difference in the end results. It would have been really helpful to have a deeper analysis of this, maybe with some t-SNE results, to better grasp how these representations work.

**Questions:**

How are the k-NN selected for both training and validation phase?

---

> ### Author Response · Authors · 2023-11-17
> **Response to Reviewer bsih**
>
> We thank the reviewer for the careful reading and valuable feedback on our submission. We are deeply encouraged that the reviewer appreciated our idea and the presentation is clear.  The acceptance of our work is particularly heartening, and we extend our sincere thanks for the insightful review. Here we provide more interpretable analysis, quantitative results and detailed descriptions of our method. We hope that our response addresses your concerns.
>
>
> **[Concern 1]** On the computational complexity issue of hypergraph construction.
>
> Thank you for bringing up this point. We acknowledge that the conventional k-NN algorithm used in hypergraph construction can be computationally intensive. However, it is crucial to note that this process, with a complexity of \\(O(n^2d)\\), is a preprocessing step rather than an inherent component of our model, and it does not impact the model's efficiency during training and testing​​.
>
> In practice, our hypergraph construction has shown acceptable preprocessing times, and we mitigate scalability concerns in two ways:
> 1. Offline Construction: The hypergraph is constructed offline and saved, allowing for a one-time computational cost that does not recur during model training or inference.
> 2. Alternative Methods: While we currently employ the k-NN algorithm, alternative methods that can group semantically similar vertices with lower complexity can be explored in future work.
>
> Regarding the handling of complex data such as high-resolution images, the computational burden does not increase with data complexity since the feature extraction is handled by pre-trained models.
>
>
> **[Concern 2]** On the interpretability of the learned representations.
>
> We thank you for your constructive feedback regarding the interpretability of our learned representations. To address this, we have conducted a t-SNE analysis to provide a visual exploration of how our HyperRep framework captures high-order correlations, which is integral to its performance.
>
> As shown in the newly included t-SNE visualization in the revised manuscript (Fig.10, Appendix L), the distinct clusters formed by the data points illustrate the efficacy of our model in learning separable and interpretable representations. Each color in the visualization represents a different category, revealing how well the HyperRep framework groups instances with high semantic similarity.
>
> The visualization highlights that representations learned with MFB loss (Fig.10 (d)) result in more distinct and cohesive clusters, compared to those relying solely on InfoNCE (Fig.10 (c)). This supports the quantitative findings that high-order correlations play a significant role in the final performance scores, demonstrating their importance in capturing complex multimodal interactions that InfoNCE alone might not fully encapsulate.
>
> Moreover, the ablation study without high-order correlation (Fig.10 (b)) falls short in terms of clustering quality, as evidenced by the more dispersed clusters and less distinct group boundaries. This visually corroborates the quantitative analysis which shows a notable drop in performance metrics when high-order correlations are omitted, underscoring their role in enhancing the discriminative power of the learned representations.
>
> Through this qualitative analysis using t-SNE and the accompanying quantitative results, we hope to have adequately addressed your concerns regarding the interpretability of the learned representations.
>
> **[Concern 3]** On quantitative results of text-to-video retrieval task.
>
> Thank you for your suggestion. In response, we have incorporated the results of two baseline methods, MCN and MIL-NCE, into Figure 9 in our revised manuscript.
>
>
> **[Concern 4]** About the selection of k-NN for training and validation.
>
> Thank you for giving us the opportunity to clarify the selection of the k-NN parameter for both the training and validation phases in our experiments. We have set the k value for the k-NN method within the hypergraph construction to 7. As detailed in Appendix J, our sensitivity analysis indicates that this parameter is not highly sensitive, allowing for a range of values to be effective. This setting is consistently applied across all datasets to ensure methodological consistency, and it is used to construct the modality hypergraphs for the entire dataset, encompassing both training and testing instances.
>
> For clustering tasks, the entire dataset is employed for both training and validation, hence the hypergraphs are constructed on the full dataset at once. For tasks such as text-to-video retrieval and action localization, which are performed in a semi-supervised manner as described in Appendix G, the hypergraph structure is also constructed using the full dataset. The only exception is that the instance hypergraph in the test set is excluded from this process. However, this does not impact the construction of the modality hypergraph, which still utilizes k-NN across the entire dataset.

---

> > ### Comment · Reviewer_bsih · 2023-11-23
> > **Thanks for the replies**
> >
> > Thanks for the replies. I have read the feedback and I maintain my review score.

---

> > > ### Author Response · Authors · 2023-11-23
> > > **Gratitude for Review Confirmation**
> > >
> > > Thank you for taking the time to consider our rebuttal. We appreciate your attention and are grateful for the affirmation of your review score.

---

### Official Review · Reviewer_yMHQ · 2023-11-01

**Soundness:** 3 good
**Presentation:** 3 good
**Contribution:** 2 fair
**Rating:** 6
**Confidence:** 4

**Summary:**

This paper presents a self-supervised representation learning method on multimodal data based on hypergraph-based learning called HyperRep by combining the strength of hypergraph-based modeling with a self-supervised multimodal fusion information bottleneck principle. The extensive experiments on four public datasets including three downstream tasks demonstrate the advantages of the proposed method, which are also validated by the comparison with the state-of-the-art approaches.

**Strengths:**

1.	The idea of using hypergraph-based self-supervised multimodal representation learning is interesting as it captures the high-order relationships in multimodal data.
2.	Compared to most multimodal learning work using semi-supervised approaches which requires additional label information, the proposed approach is employing self-supervised learning. The derived hypergraph attention module and propagation is a good extension from current approaches.
3.	The author also introduced multimodal fusion information bottleneck (MFB) principle, in order to maximize the mutual information between the instance and each modality. The corresponding upper bound and lower bound are derived.
4.    The experimental evaluation is detailed and extensive.

**Weaknesses:**

1. Overall, while the idea is interesting, the scope is relatively narrow as it combines self-supervised learning with multimodal representation learning based on hypergraph. However, hypergraph attention network for multimodal learning has been explored in
https://openaccess.thecvf.com/content_CVPR_2020/papers/Kim_Hypergraph_Attention_Networks_for_Multimodal_Learning_CVPR_2020_paper.pdf
Compared to this CVPR 2020 paper, the only novel part for the manuscript seems to be adding self-supervised learning on top of it. However, that novelty is relatively small.

2. The performance gain appears to be incremental as the paper serves as an extension to recent work.

**Questions:**

1 The author needs to distinguish the work better with the previous work especially many module presented in this paper such as hypergraph attention module has been published before. So a reference shoud be given and explain the difference if there is any.

2.While the lower bound and upper bound are derived, it is important to show how to leverage these bounds for optimization.

3.The contribution of the paper is not very clear.

---

> ### Author Response · Authors · 2023-11-17
> **Response to Reviewer yMHQ (1)**
>
> We thank the reviewer for the careful reading and valuable feedback on our submission. We are deeply encouraged by the reviewer's appreciation of our idea's originality and the detailed, extensive nature of our experiments. The acceptance of our work is particularly heartening, and we extend our sincere thanks for the insightful review. In the following sections, we provide more detailed descriptions and illustrations to further elucidate the novelty, contribution, and effectiveness of our method. We hope that our response comprehensively addresses your concerns.
>
> **[Concern 1]** On the novelty issue against HAN[1].
>
> Hypergraph Attention Networks (HAN) [1] are specialized for the Visual Question Answering (VQA) task, employing hypergraphs to create co-attention maps between multimodal inputs and considering structural similarity to embody high-level semantic relationships within symbolic information. In our scenario, where the extraction of multi-level high-order correlations and information propagation from vertices to hyperedges for downstream tasks is essential, the methods employed by HAN prove to be inadequate. The random walks sampling method and graph-matching techniques utilized by HAN cannot achieve these objectives, as HAN only regards hyperedges as alternatives to subgraphs without specific semantic meaning.
>
> HyperRep, our proposed method, addresses these limitations by constructing modality hyperedges using the K-NN algorithm, thereby capturing high-order semantic correlations within modalities. Instance hyperedges are naturally constructed to extract cross-modal correlations within instances. Additionally, the attention mechanism in HyperRep aims for accurate information propagation instead of just extracting a joint representation across modalities, further differentiating it from HAN.
>
> Despite the self-supervised learning part, the discrepancies between HAN and HyperRep in terms of hyperedges' nature, attention mechanisms, construction methods, and types of hyperedges (with HyperRep considering cross-modality hyperedges while HAN neglects) underline the significant differences between the two. Though both approaches utilize hypergraphs and attention mechanisms, their underlying philosophies and methodologies are task-specific and markedly distinct. HAN's symbolic graph construction and graph-matching method are not directly applicable to our scenario, thereby emphasizing the novelty of HyperRep.
>
> [1] Hypergraph Attention Networks for Multimodal Learning, CVPR 2020.
>
> **[Concern 2]** On the performance gain.
>
> We appreciate your observation and would like to address the concern regarding the perceived incremental nature of our work. HyperRep represents a substantial leap forward rather than a mere extension of existing methodologies. Our approach introduces a novel self-supervised learning framework within the realm of hypergraphs, which distinctly captures high-order correlations across multimodal data—a significant departure from prior models.
>
> The performance gains of HyperRep are far from incremental. Our model demonstrates a marked improvement over existing state-of-the-art methods. For instance, in clustering tasks, HyperRep leads by significant margins such as 8.2%, 6.9%, and 16.7% in Acc, NMI, and ARI metrics respectively on the AVE dataset​​. This trend of outperformance is consistent across various datasets and tasks, as shown in our extensive experimental results. In text-to-video retrieval tasks, our method observes improvements of 10.5%, 4.4%, and 10.4% in Recall@1, Recall@5, and Recall@10 respectively, compared to the next best method​​. Such substantial gains underscore the effectiveness of our model's ability to learn nuanced, cross-modal representations that are deeply grounded in the data's semantic structure.
>
> Furthermore, our approach to constructing hyperedges that capture both inter- and intra-modality semantic correlations, coupled with the introduction of the MFB principle to constrain the solution space, is unique and instrumental to our model's performance. The observed improvements are indicative of HyperRep’s enhanced capacity to understand the underlying clustering structure and the genuine similarities among instances, as evidenced by the strong results.
>
> In essence, the development and application of the MFB loss function, which optimizes for shared information across modalities, represent a key innovation that substantially enhances the model's performance. These are not minor incremental updates; they are fundamental contributions that redefine the approach to multimodal representation learning.
>
> We believe these points collectively illustrate that HyperRep is a significant and distinct contribution to the field, offering far-reaching implications for multimodal learning research.

---

> > ### Comment · Reviewer_yMHQ · 2023-11-22
> > **Final comments**
> >
> > Thanks for the detailed feedback from the authors.
> >
> > I have read via the feedback and I maintain my review score.

---

> > > ### Author Response · Authors · 2023-11-22
> > > **Appreciation for Review Confirmation**
> > >
> > > Thank you for taking the time to consider our feedback. We appreciate your attention to detail and are grateful for the affirmation of your review score.

---

> ### Author Response · Authors · 2023-11-17
> **Response to Reviewer yMHQ (2)**
>
> **[Concern 3]** About leaverage the lower bound and upper bound of the MFB for optimization.
>
> We appreciate the opportunity to clarify the optimization process concerning the bounds of the Multimodal Fusion Information Bottleneck (MFB). As delineated in our manuscript, the MFB principle introduced in Eq.12 is designed to balance maximizing mutual information between the instance and each modality while minimizing mutual information between the instance and the collective modalities. To facilitate the optimization of this principle, especially when the probability distribution is unknown, we derived tractable estimations for both upper and lower bounds of mutual information as outlined in Proposition 1.
>
> Concretely, we sought the lower bound for the first term in Eq.12, as it is negated, and the upper bound for the second term. These bounds are crucial for transforming the intractable mutual information components into a computable loss function suitable for gradient-based optimization methods. The upper bound of Eq.12 is thus formulated in Eq.13, which we refer to as the MFB loss. This loss function encapsulates the InfoNCE loss as the lower bound for the mutual information, effectively allowing us to optimize the MFB principle using backpropagation, as the InfoNCE loss provides a gradient-friendly surrogate objective.
>
> The detailed derivation of the MFB loss, including the application of Proposition 1, which presents the mutual information estimations, is meticulously documented in Appendix C of the manuscript. This derivation ensures that the optimization process aligns with the theoretical underpinnings of MFB, guiding the learning process to focus on shared multimodal information, thereby effectively constraining the solution space and directing the model’s attention towards inter-modality shared information, as intended by the MFB principle.
>
> We trust this explanation elucidates the optimization strategy for the MFB principle and its implementation within our model. We are grateful for your feedback, which has allowed us to better communicate the theoretical and practical aspects of our approach.
>
> **[Concern 4]** On the contribution issue.
>
> We appreciate your feedback and the opportunity to more explicitly clarify the contributions of our paper, HyperRep.
>
> HyperRep introduces a novel hypergraph-based framework for capturing high-order correlations across and within multiple modalities, which significantly diverges from existing methods in the literature. This framework offers a comprehensive approach to model the inter- and intra-modality high-order correlations in multimodal data. A key element of our approach is the proposed Multimodal Fusion Bottleneck (MFB) principle, a unique mechanism designed to optimize information fusion by balancing mutual information within and across modalities. This principle is rigorously defined, with detailed proofs, derivations, and calculations.
>
> The effectiveness of HyperRep is further evidenced through extensive experiments. Contrary to incremental advancements, our model demonstrates significant improvements across several benchmarks. In clustering tasks, for example, HyperRep outperforms second-best methods by substantial margins, achieving an average improvement of 6.45% across three datasets. Our rigorous ablation studies also highlight the significant impact of each component, including the high-order correlations and the MFB loss function.
>
> Additionally, the adaptability and scalability of HyperRep have been proven across various downstream tasks, such as text-to-video retrieval and temporal action localization. Complementing these, we conducted comprehensive analyses, including sensitivity analysis, computational complexity analysis, convergence analysis, and quantitative analysis, all of which underscore the robustness and versatility of our method.
>
> We trust that these points adequately demonstrate the significant contributions of HyperRep to the advancement of multimodal representation learning.

---

### Author Response · Authors · 2023-11-21
**Follow-up on Recent Rebuttal Submission and Further Discussion**

Dear reviewers,

I hope this message finds you well. I wanted to extend a gentle follow-up on the rebuttal we submitted a few days ago concerning our manuscript. We are eager to hear your thoughts and understand if the additional clarifications and explanations provided have addressed the concerns you raised. We deeply value your insights and are fully open to further discussion should there be any outstanding issues or new queries. We appreciate the time and effort you invested in this review process and look forward to your valuable feedback.

---

### Comment · Area_Chair_fRft · 2023-11-22
**Let's have more discussion with authors**

Dear reviewers,

Your interaction with the authors on this work is highly appreciated.

The author-reviewer discussion period is closing at the end of Wednesday Nov 22nd (AOE). Let's take this remaining time to have more discussions with the authors on their responses to your reviews. Should you have any further opinions, comments or questions, please let the authors know asap and this will allow the authors to address them.

Kind regards, AC

---

### Meta-Review · Area_Chair_fRft · 2023-12-09

**Metareview:**

Based on the submission, reviews, and author feedback, the main points that have been raised are summarised as follows.

Strengths:

1. The idea is interesting, and the proposed method employs self-supervised learning.
2. The hypergraph attention module and propagation is a good extension from existing work.
3. Experimental evaluation is detailed and extensive. There is a consistent improvement in performance.
4. The paper is well written and nicely structured.

Issues:

1. Need to distinguish the work better with the previous work especially many module presented in this paper.
2. Some performance gain appears to be incremental and more comparison with recent methods is needed.
3. Need detailed analysis of the interpretability of the learning representations and the role of InfoNCE in this method.
4. Motivation and the presentation of MFB could be clarified.

The authors provided feedback to address the above raised issues. Most reviewers have read the feedback and responded to the authors.
After reading this submission, AC agrees that this work has its merits and demonstrates performance improvement. Meanwhile, overall this work is not sufficiently novel when compared with the related literature (e.g., using hypergraph to model multimodal data); the MFB part needs to be more clearly explained and more thoroughly justified (including experimental justification since the difference between ``L_{InfoNCE} only'' and ''full model'' in Table 3 is small, especially for YouCook2 dataset); and the proofs related to MFB in Appendix needs to be made clearer and further examined (e.g., the step from Eq.(20) to Eq.(21) on page 13 and the step from Eq.(27) to Eq.(28) on page 14). AC discussed this work with SAC.

**Justification For Why Not Higher Score:**

This work has its merits and demonstrates performance improvement. Meanwhile, overall this work is not sufficiently novel when compared with the related literature; the MFB part needs to be more clearly explained and more thoroughly justified; and the derivations in Appendix needs to be made clearer and further examined. Therefore, a higher score is not recommended.

**Justification For Why Not Lower Score:**

N/A

---

### Decision · Program_Chairs · 2024-01-16

Reject